# Clustering of health behaviors among Japanese adults and their association with socio-demographics and happiness

Miho Satoh ⓘ *

Department of Fundamental Nursing, Yokohama City University, Yokohama, Kanagawa, Japan

* miho.sth@gmail.com

## Abstract

### Background

Scholars have associated happiness with health behaviors, which co-occur and exert synergistic effects. Therefore, identifying clusters of health behaviors and their effect on happiness can contribute to the development of strategies for promoting happiness and improving health behaviors.

### Aim

This study aimed to examine clusters of health behaviors and their associations with socio-demographics and with happiness among Japanese adults.

### Methods

This study used data from the Japanese Household Panel Survey and the Keio Household Panel Survey. Questionnaires were distributed to 4,993 households out of which 1,554 responses were analyzed (participants aged 27−65 years). The survey included health behaviors (alcohol consumption, smoking, vegetable and fruit consumption, breakfast-eating habits, and physical activities), happiness, and socio-demographics. Latent class analysis was conducted to identify the clusters, whereas latent regression was employed to investigate socio-demographics related to the clusters. Logistic regression analysis was conducted to examine the relation between happiness and the clusters.

### Results

Two health behavior clusters were identified: Cluster 1 (low substance use and good dietary habit; n = 817; 52.3%) and Cluster 2 (high alcohol, poor nutrition, and inactive; n = 737; 47.7%). Latent regression analysis indicated that all socio-demographics, barring socioeconomic status, were significantly associated with the clusters. The "low substance use and good dietary habit" cluster was significantly related with higher odds of happiness (odds ratio = 1.425, 95% confidence interval = 1.146−1.772, $p$ = 0.001).

**Data Availability Statement:** The third-party data set (Approval No. 1322: Data ID 156-JHPS/KHPS2004-2018) supporting the conclusions of this article is available from the Panel Data Research Center at Keio University with the

approval of this organization (https://www.pdrc.
keio.ac.jp/en) for researchers who meet the criteria
for access to confidential data. The authors confirm
that they did not have any special access privileges
that others would not have. The authors also
confirm that others would be able to access these
data in the same manner as themselves.

**Funding:** Unfunded studies.

**Competing interests:** No author has competing
interests.

## Conclusions

This study identified health behavior clusters among Japanese adults and established the
association between the "low substance use and good dietary habit" cluster and high levels
of happiness. However, the causality of the relationship between health behavior and happi-
ness remains unclear, which highlights the need for further research to elucidate the under-
lying mechanisms.

## Introduction

Happiness, which is a key facet of well-being, productivity, and quality of life, has received
much scholarly attention worldwide [1–8]. Conceptually speaking, happiness refers to the
overall appreciation of life and to a subjective state of mind [9]. It has been frequently deemed
as an ultimate life goal [5]. According to Diener [10], happiness corresponds to subjective
well-being and can be considered a global term for various types of positive and negative evalu-
ations of people about their lives. In general, happiness is a concept related to well-being,
which can be comprehensively assessed in terms of its positive and negative aspects; therefore,
it may serve as an effective indicator for the measurement of overall satisfaction with life
situations.

Studies on happiness have flourished as a field in economics with several researchers exam-
ining the association between happiness, socioeconomic status (SES), social capital, and suc-
cess in life [11–13]. Numerous studies on health have identified factors associated with
happiness, such as morbidity and mortality [4, 14], absence of disease [4], and subjective and
psychological health [15].

Scholars have found that healthy lifestyles, such as being physically active [16, 17], consum-
ing vegetables and fruits [18–20], not smoking [21, 22], and consuming alcohol in moderation
[23–25], are significantly associated with happiness [26]. Thus, further engagement in health
behaviors could enhance happiness [27]. Nonetheless, lifestyles are composed of multiple
health behaviors. Thus, many studies have investigated the relationship between health behav-
iors and physical or psychological health status in isolation.

A few studies in Europe and the United States have found that multiple health behaviors
tend to occur interrelatedly and that the degree of health consciousness exerts a cascading
effect on all types of health behaviors [28–30]. In other words, one health behavior can trigger
a chain reaction of other health behaviors, whereas the opposite is also true. Moreover, the
simultaneous occurrence of multiple health behaviors has been attributed to various mecha-
nisms, such as the commonality of multiple factors associated with such behaviors, and the
fact that one behavior may be a coping strategy for another [31]. Multiple health behaviors
exert synergistic effects. Therefore, research on the clustering of multiple health behaviors and
their relationship with health outcomes is increasing [28, 30, 32–36].

Furthermore, scholars have analyzed the synergistic effects of multiple health behaviors on
happiness [23, 37]. Specifically, the research on the association between multiple health behav-
iors and happiness has focused on younger populations of adolescents and college students
[33–35]. However, less is known about multiple health behaviors across the full range of adult
age population. Specifically, studies on this topic are lacking in Japan.

Interventions that target only one health behavior can lead to consequences for improving
other co-occurring health behaviors, which enhances happiness accordingly. Against this
background, altering health behavior patterns could be an effective means for increasing

happiness. Moreover, identifying the existing clusters of health behaviors and the characteristics of people belonging to these clusters would be beneficial to the formulation of intervention strategies for changing health behavior patterns.

This study aimed to investigate clusters of health behaviors, the socio-demographics related to these clusters, and the association between clusters of health behavior and happiness.

## Materials and methods

The study used the data from the Japanese Household Panel Survey (JHPS) and the Keio Household Panel Survey (KHPS), which were conducted by Keio University. The surveys provide representative data on Japanese households from two separate sources.

Ethical approval was not required because this study used only secondary analysis [38]. Consent was obtained from all participants in the JHPS/KHPS regarding the purpose of the research, potential use of their data, data anonymity, strict protection of individual data, and secondary use of data by the Keio University Panel Data Research Center.

The KHPS and JHPS have been conducted annually since 2004 and 2009 and have conducted surveys on 4,005 and 4000 households, respectively.

A stratified two-stage sampling method for the data from both surveys and for recruiting the participants was used. The KHPS and JHPS featured male and female adults aged between 20–69 years and 20 years and above, respectively. The demographics of the respondents of both surveys are representative of Japanese households nationwide. Moreover, the two surveys have no overlapping participants. The questionnaires used in these surveys include items on place of residence, basic demographic data (e.g., year of birth, level of education, and height), household structure, employment status, school attendance, daily schedule, long-term care status, social insurance, health status, health behaviors, and household economic status.

Data were collected primarily from the 2017 waves of both surveys; however, the 2004 and 2005 waves of the KHPS and the 2009 wave of the JHPS were partially derived to collect basic demographic data.

### Variables

**Health behaviors.**    The health behaviors emphasized by previous studies are smoking, alcohol consumption, diet, and exercise [28–30, 39–41] with recent studies focusing on the relationship between vegetable and fruit intake and physical and psychological health [18–20, 28–30]. In addition, other studies analyzed breakfast habits and their association with physical and psychological health [28, 29].

Thus, the current study employed alcohol consumption, smoking, vegetable and fruit consumption, breakfast-eating habit, and physical activity as health behavior variables [39–41]. After a literature review [39–42], each variable takes a value of 0 or 1 for high or low risk, respectively.

Alcohol consumption was assessed using two questions. The first was "How frequently do you consume alcohol?" This item was rated as Never; A few times per month; 1–2 times per week; More than three times a week; and every day. The second question was "What is your average alcohol consumption in day when converted to sake?" which is rated as less than 180 ml; 180–360 ml; 360–540 ml; 540–720 ml; 720–900 ml; and More than 900 ml. Sake contains 23 g of alcohol per 180 ml. Thus, responses were coded as 0 = More than 23 g of alcohol daily and 1 = No alcohol consumption or less than 23 g of alcohol daily.

Smoking status was assessed using the question "Do you smoke cigarettes?" with the following responses: I smoke; I smoke sometimes; I have quit smoking; and Never. The World Health Organization (WHO) recommends refraining from smoking tobacco to maintain good

health [43, 44]. Responses were coded as 1 for current non-smokers (responded "I have quit smoking" or "Never") and 0 for current smokers (responded "I smoke" or "I smoke sometimes").

The consumption of vegetables and fruits was assessed using one question with vegetables and fruits as options: "How often did you eat the foods listed below in the last one month?" The responses were: Three times per day; Twice per day, Once per day; 4–6 times per week, 2–3 times per week; Once per week, 1–3 times per month; and Never. Responses were coded as 0 for "less than two times daily" and 1 for "more than two times daily." Consumption was measured with reference to the recommendation by the Ministry of Health, Labor and Welfare to eat at least two meals per day with side dishes including vegetables and fruits or an intake of at least two servings per day of fruits and vegetables [41, 42].

Breakfast-eating habit was assessed using the question: "How often do you eat breakfast?" with the following responses: Almost every day; Skip 2–3 times/week; Skip 4–5 times/week; and Skip almost every day. Responses were coded 0 for "less than 4 days/week" and 1 for "more than 4 days/week."

Physical activity was assessed using the question: "How many days do you exercise each week in which you sweat?" with the following responses: 0 days; 1 day; 2 days; 3 days; 4 days; 5 days; 6 days; and 7 days. Responses were coded as 0 for "less than 2 days" and 1 for "more than 2 days."

**Happiness.** The respondents rated perceived happiness during the past year on a scale from 0 = "Having no feeling of happiness at all" to 10 = "Having a feeling of complete happiness." This single-item scale was used in the National Survey of Lifestyle Preference by the Japanese government [45]. The distribution of happiness in the sample of this study was bimodal with a median of 6. Thus, it was divided into two groups: high (scores of 7 or higher) and low (scores of 6 or lower) levels of happiness.

**Psychological health.** As psychological health can influence health behaviors and happiness, the depression level of the participants was used as the control variable. Psychological health was measured using the Japanese version of the 12-item General Health Questionnaire (GHQ) [46, 47]. Items were rated using a four-point Likert-type scale (1 = Less than usual; 2 = No more than usual, 3 = Relatively more than usual, and 4 = Much more than usual) and were converted into binary values (i.e., 0, 0, 1, and 1), where high total scores indicated more severe psychological distress. The cut-off point of the GHQ was set at 2/3 [46, 47]; those scoring 3 points or higher were considered to indicate depression.

**Socio-demographics.** The following socio-demographics were assessed: sex, year of birth, marital status (i.e., married or single), living with family (i.e., yes or no), level of education (i.e., junior or senior high school graduate, junior college graduate, college degree, or graduate degree), SES (i.e., expressed in increments of 10,000 JPY; 1 USD was approximately 113 JPY at the time of the survey), and employment status (i.e., regular: full-time with a contract for regular employment; non-regular: part-time or temporary contract; self-employed; or non-employed: those who are neither workers nor unemployed, such as full-time housewives, students, or retirees) [48, 49]. Based on age, the participants were classified into three groups: younger (27–44 years), middle (45–55 years), and older (56–65 years). Regarding SES, the participants described their disposable income per household, such that the responses were divided into three groups based on the quartile: lower (<2,150 K, where K is 1,000 yen), middle (2,150 K to<4,040 K), and higher (≥4,040 K).

## Data analysis

The same questionnaires were used for both surveys, which were, thus, conducted at the same time each year (January). The two datasets were combined for data analysis as they have no overlapping respondents.

Latent class analysis (LCA) was conducted to identify the underlying patterns (or classes) of health risk behaviors: alcohol consumption, smoking, vegetable consumption, fruit consumption, breakfast-eating habit, and physical activity. A sequence of LCA models was estimated by specifying between two and six classes to determine the number of classes that best represent the patterns of health behaviors. Bayesian Information Criterion and Akaike Information Criteria were used as goodness-of-fit indices, were generated for each model [50], and used to interpret the clusters to determine the number of classes.

Latent class regression was used to explore the association between health behavior classes and socio-demographic variables, where GHQ-12 was used as the control variable. Finally, multiple regression analysis was conducted to elucidate the association between happiness and health behaviors (all socio-demographic variables were used as control variables).

All statistical analyses were conducted using JMP 15.0 and SPSS Statistics 26.0 for Mac. Statistical significance was set to $p < 0.05$ (two-tailed).

## Results

In 2017, the KHPS questionnaire was distributed to 2,945 households, out of which 2,729 responses were received (response rate: 92.7%). In the same year, the JHPS questionnaire was distributed to 2,048 households out of which 1,882 responses were received (response rate: 91.9%). Participants with missing data on the socio-demographic attributes (employment status and academic background) and health behaviors were excluded. The average age of the participants with missing data was 53.98 ± 11.23 years contrary to 49.84 ± 9.16 years of the analyzed sample ($t = -11.60$, $p < 0.000$), whereas 59.2% contrary to 59.5% were male in the analyzed sample ($\chi^2 = 114.18$, $p < 0.000$). In total, the current study analyzed the responses of 1,554 completed questionnaires (valid response rate: 31.1%).

### Characteristics of the participants

Table 1 presents the characteristics of the participants. The mean age was 49.84 ± 9.16 years; 59.5% were male; 52.6% were regular employees; 77.3% were married; and 34.6% had depression.

Table 2 provides the prevalence of health behaviors. In total, 76.9% of the participants were non-smokers, whereas 38.5% reported no alcohol consumption or <23 g of alcohol consumption per day. Furthermore, 46.2% and 7.1% consumed vegetables and fruits, respectively, more than two times daily; 75.3% reported eating breakfast every day, whereas 17.8% reported engaging in physical activities for more than two times per week.

A two-class model was favored according to the goodness of fit and interpretability of the clusters. Fig 1 depicts the conditional probabilities of the two clusters. Cluster 1 (n = 817; 52.3%) was characterized by the lowest probability for smoking, moderately low alcohol consumption, highest vegetable intake, and regular breakfast intake and was described as "low substance use and good dietary habit." Cluster 2 (n = 737; 47.7%) was described as "high alcohol, poor nutrition, and inactive," with participants characterized by high levels of alcohol consumption, low vegetable intake, very low fruit intake, and very low physical activity. The participants in Cluster 2 displayed lower levels of breakfast-eating habit and physical activity than those in Cluster 1.

Using the "high alcohol, poor nutrition, and inactive" cluster as reference, the results of the latent regression analysis demonstrated that males (odds ratio [OR] = 0.379, 95% confidence interval [CI] = 0.290−0.497, $p = 0.000$), regular employees (OR = 0.540, 95% CI = 0.341−0.857, $p = 0.009$), non-regular employees (OR = 0.576, 95% CI = 0.368−0.903, $p = 0.016$), and self-employed individuals (OR = 0.539, 95% CI = 0.322−0.901, $p = 0.018$) displayed lower odds of

**Table 1. Characteristics of the participants (N = 1,554).**

|  | n | % |  |
|---|---|---|---|
| Sex |  |  |  |
| Male | 924 | 59.5 |  |
| Female | 630 | 40.5 |  |
| Age (Years; Mean ± SD) | 49.84 ± 9.16 | | 27–65 |
| Younger (27–44) | 230 | 14.8 |  |
| Middle (45–55) | 834 | 53.7 |  |
| Older (56–65) | 490 | 31.5 |  |
| Marital status |  |  |  |
| Married | 1,202 | 77.3 |  |
| Single | 352 | 22.7 |  |
| Living with family |  |  |  |
| Yes | 1,408 | 90.6 |  |
| No | 146 | 9.4 |  |
| Level of education |  |  |  |
| Junior/senior high school or junior college graduate | 979 | 63.0 |  |
| College or graduate degree | 575 | 37.0 |  |
| SES |  |  |  |
| Lower (<2,150 K) | 386 | 24.8 |  |
| Middle (2,150 K to <4,040 K) | 775 | 49.9 |  |
| Higher (≥4,040 K) | 393 | 25.3 |  |
| Employment status |  |  |  |
| Regular | 817 | 52.6 |  |
| Non-regular | 408 | 26.3 |  |
| Self-employed | 197 | 12.7 |  |
| Non-employed | 132 | 8.5 |  |
| Depression |  |  |  |
| cYes | 538 | 34.6 |  |
| No | 1016 | 65.4 |  |

SD: Standard deviation; SES: Socioeconomic status

K: 1,000 yen

belonging to the "low substance use and good dietary habit" cluster. Meanwhile, participants who were married (OR = 1.521, 95% CI = 1.130−2.047, $p$ = 0.006), with high levels of education (OR = 2.333, 95% CI = 1.842−2.955, $p$ = 0.000), and with high levels of SES (OR = 1.483, 95% CI = 1.083−2.030, $p$ = 0.014) had higher odds of belonging to the "low substance use and good dietary habit" cluster. Moreover, those with depression had lower odds of belonging to the "low substance use and good dietary habit" cluster (Table 3).

Furthermore, Table 4 indicates that individuals under the "low substance use and good dietary habit" cluster exhibited significantly higher odds of feeling happiness (OR = 1.266, 95% CI = 1.010−1.587, $p$ = 0.041).

## Discussion

The study identified two clusters of health behaviors, which were associated with age, sex, marital status, family status, level of education, and employment status. In addition, the results demonstrated that high levels of happiness were significantly associated with health behaviors.

**Table 2. Prevalence of health behaviors (N = 1,554).**

|  | n | % |
|---|---|---|
| Smoking status |  |  |
| Smokers | 359 | 23.1 |
| Non-smokers | 1,195 | 76.9 |
| Alcohol consumption |  |  |
| ≥23 g | 956 | 61.5 |
| No alcohol consumption or <23 g | 598 | 38.5 |
| Vegetable intake |  |  |
| More than two times daily | 718 | 46.2 |
| Less than two times daily | 836 | 53.8 |
| Fruit intake |  |  |
| More than two times daily | 110 | 7.1 |
| Less than two times daily | 1444 | 92.9 |
| Breakfast-eating habit |  |  |
| More than four days/week | 1,170 | 75.3 |
| Less than four days/week | 384 | 24.7 |
| Physical activity |  |  |
| ≥Two days per week | 276 | 17.8 |
| <Two days per week | 1278 | 82.2 |

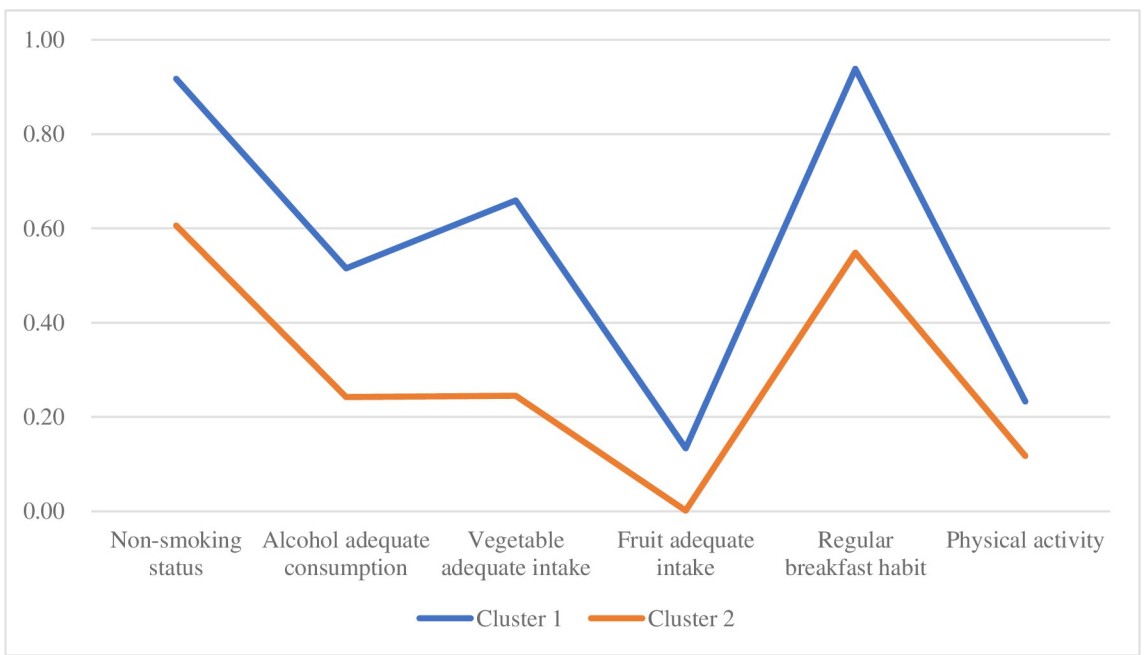

| Ns | Non-smoking status | Adequate alcohol consumption | Adequate vegetable intake | Adequate fruit intake | Regular breakfast eating habit | Physical activity |
|---|---|---|---|---|---|---|
| Cluster 1 | 777 | 472 | 565 | 110 | 803 | 220 |
| Cluster 2 | 418 | 126 | 153 | 0 | 367 | 56 |

**Fig 1. Conditional item probabilities of membership.**

**Table 3. Association between the "low substance use and good dietary habit" cluster and socio-demographics.**

| | OR | 95% CI | | | p-Value |
|---|---|---|---|---|---|
| Male (female = 0) | 0.379 | 0.290 | – | 0.497 | 0.000 |
| Age (younger = 0) | | | | | |
| Middle (45–55) | 0.873 | 0.636 | – | 1.199 | 0.402 |
| Older (56–65) | 1.509 | 1.062 | – | 2.143 | 0.022 |
| Married (Single = 0) | 1.521 | 1.130 | – | 2.047 | 0.006 |
| Living with family (No = 0) | 1.463 | 0.955 | – | 2.242 | 0.081 |
| College or graduate degree (Junior or senior high school, or junior college graduate = 0) | 2.333 | 1.842 | – | 2.955 | 0.000 |
| SES (Lower = 0) | | | | | |
| Middle (2,150 K to <4,040 K) | 1.262 | 0.968 | – | 1.646 | 0.085 |
| Higher (≥4,040 K) | 1.483 | 1.083 | – | 2.030 | 0.014 |
| Employment status | | | | | |
| (Non-employed = 0) | | | | | |
| Regular | 0.540 | 0.341 | – | 0.857 | 0.009 |
| Non-regular | 0.576 | 0.368 | – | 0.903 | 0.016 |
| Self-employed | 0.539 | 0.322 | – | 0.901 | 0.018 |
| Depression (No = 0) | 0.789 | 0.631 | – | 0.986 | 0.038 |
| Nagelkerke's $R^2$ | 0.147 | | | | |

Reference: Cluster 2: High alcohol, poor nutrition, and inactive

CI: Confidence interval; SES: Socioeconomic status; OR: Odds ratio

**Table 4. Association between clusters and happiness.**

| | OR | 95% CI | | | p-Value |
|---|---|---|---|---|---|
| Male (Female = 0) | 0.601 | 0.454 | – | 0.796 | 0.000 |
| Age (Younger = 0) | | | | | |
| Middle (45–55) | 0.884 | 0.638 | – | 1.224 | 0.457 |
| Older (56–65) | 0.670 | 0.467 | – | 0.960 | 0.029 |
| Married (Single = 0) | 2.255 | 1.651 | – | 3.080 | 0.000 |
| Living with family (No = 0) | 1.138 | 0.726 | – | 1.784 | 0.572 |
| College or graduate degree (Junior or senior high school, or junior college graduate = 0) | 1.636 | 1.288 | – | 2.077 | 0.000 |
| SES (Lower = 0) | | | | | |
| Middle (2,150 K to <4,040 K) | 1.242 | 0.948 | – | 1.627 | 0.116 |
| Higher (≥4,040 K) | 2.000 | 1.451 | – | 2.755 | 0.000 |
| Employment status | | | | | |
| (Non-employed = 0) | | | | | |
| Regular | 1.734 | 1.103 | – | 2.727 | 0.017 |
| Non-regular | 1.611 | 1.045 | – | 2.483 | 0.031 |
| Self-employed | 1.820 | 1.097 | – | 3.022 | 0.021 |
| "Low substance use and good dietary habit" cluster ("High alcohol, poor nutrition, and inactive" cluster = 0) | 1.266 | 1.010 | – | 1.587 | 0.041 |
| Depression (No = 0) | 0.296 | 0.235 | – | 0.374 | 0.000 |
| Nagelkerke's $R^2$ | 0.186 | | | | |

Reference: Lower level of happiness group

Cluster 1: Low substance use and good dietary habit; Cluster 2: High alcohol, poor nutrition, and inactive; CI: Confidence interval; SES: Socioeconomic status; OR = odds ratio

Based on the distribution of health behaviors, participants in the "low substance use and good dietary habit" cluster exhibited low health risk behaviors than those in the "low substance use and good dietary habit" cluster. Having breakfast regularly may be associated with high vegetable consumption [51–53]. Previous studies have suggested that those who show high health awareness are likely to have some healthy behaviors simultaneously [37, 54–57]. Thus, the "low substance use and good dietary habit" cluster, comprising individuals who are non-smokers, consume moderate amounts of alcohol, eat vegetables nearly every day, eat breakfast nearly every day, even though with low fruit intake, and are physically inactive, was considered a health behavior pattern. However, individuals in this cluster that engaged in "health living" did not embrace all health recommendations equally. Previous studies added moderate engagement in physical activities to the abovementioned health behaviors [28–30]. On the other hand, the present study did not observe such pattern. In Japan, 31.8% of men and 25.5% of women are engaged in the habit of exercising (i.e., 30 min of exercise at least twice a week continuously throughout the year) [58], which is a low percentage overall. Takamiya et al. [58] conducted a retrospective study and demonstrated that, indeed, adult men and women in Japan have been unable to acquire exercise habits, such that the need for effective interventions that aim to address this issue is urgent [58]. In addition, less than 40% of adults with positive self-rated health have established exercise habits [59]. Fruit consumption has remained low among Japanese individuals [42], though positive effect of physical and psychological health of fruit consumption are widely acknowledged [18–20, 28–30]. Especially, among Japanese people in their 20s–50s, more than 50% had no regular fruit consumption [42]. Thus, the current study assumes that the results regarding physical activity and fruit consumption reflected the general characteristics of the Japanese.

The "high alcohol, poor nutrition, and inactive" cluster, which comprises high levels of alcohol consumption, very low fruit intake, slightly low levels of breakfast-eating habit, and very low physical activity, was considered an unhealthy behavioral pattern. The co-occurrence of these behaviors is congruent with previous research [28, 32, 60, 61]. The "high alcohol, poor nutrition, and inactive" cluster also exhibited higher current smoking habit and lower vegetable intake than those in the "low substance use and good dietary habit" cluster. Research demonstrated that health risk behaviors frequently do not occur in isolation. Instead, they are likely to co-occur or cluster. In previous studies on the Japanese population, several unhealthy habits tended to occur simultaneously [62–64]. In addition, the National Health and Nutrition Survey suggests that people with one health risk behavior may have other health risk behaviors [42]. Thus, the current study exhibits consistency with past research given that the "high alcohol, poor nutrition, and inactive" cluster displayed various health risk behaviors that occurred simultaneously. The combination of multiple inadequate health habits could increase their undesirable effects on health. Hence, developing interventions that focus on improving not only individual health habits but also habits that may lead to compounding effects is necessary.

These clusters of health behaviors point to significant relationships with socio-demographic variables. In particular, older people tended to adopt healthier behaviors in this study. As people become older, they develop increased health awareness and endeavor to engage in healthy behaviors and lifestyles [32, 65]. Japanese people in their 50s–60s tend to display increased levels of health awareness and adopt healthier lifestyles compared with the people in their 30s–40s [42, 66]. Younger people possess low levels of awareness because they lack subjective experience of chronic or other diseases [67, 68]. Previous studies have proposed that taking measures aimed at changing multiple health behaviors is important to reduce the risk of noncommunicable diseases (e.g., diabetes, hypertension, or cardiovascular diseases) [69, 70]. Therefore, developing and promoting health awareness-related initiatives among Japanese youth may be necessary to encourage engagement in health-promoting behaviors, which may exert a significant impact on their future health status.

Additionally, other surveys have found that people in their 30s–40s tend to face time constraints and are less aware of their health due to increased roles and responsibilities in family and work life [42, 71]. That could explain the lack of association between younger age group and the "low substance use and good dietary habit" cluster, and it might also indicate that employed people were more likely to be in the "high alcohol, poor nutrition, and inactive" cluster. The present study demonstrated that the employed Japanese population was more prone to unhealthy behaviors. A large proportion of the insured, working adults failed to meet the recommendations for health behaviors in the US [72]. Other studies have also reported that many employed people tend to observe unhealthy lifestyles, because they are extremely busy with work and fail to pay attention to their health habits [42, 73, 74]. Moreover, work-related stress may influence engagement in health risk behaviors, such as alcohol consumption [75, 76]. The work culture may also lead employed individuals to frequently go out for drinks to interact socially and professionally with bosses and colleagues after work [77, 78]. Hence, employed individuals in Japan may be more likely to adopt unhealthy behaviors, which highlights the potential need for a comprehensive analysis of the work lives of Japanese workers to improve health behavior patterns.

Prior research has demonstrated that not being married was associated with an unhealthy lifestyle [56, 79, 80]. Meanwhile, married individuals exhibited an increased tendency to engage in health behaviors for various reasons, such as having acquired close social support to encourage engagement in such behaviors and having established health behaviors through the influence of partners [28, 57, 81, 82]. The current study produced findings similar to evidence from past research.

According to previous studies, high levels of education may be significantly associated with health behaviors [30, 83], which is similar to the result obtained by the current study. In Japan, the level of education is related to health behaviors, where high levels of education are associated with more physical exercise routines [84], lower problem drinking [85], and lower obesity [86]. In addition, the current study confirmed the association between multiple healthy behaviors and the higher SES. This result was also congruent with literature [80, 87]. Scholars have speculated that this finding is due to the fact that the higher the level of education or the higher income, the more knowledge or resources people utilize for health [88, 89]. Therefore, they are more likely to engage in health-promotion activities and also gain access to useful resources that improve health. This indicates the need to examine the content of education in schools to ensure that young people, regardless of their SES, gain appropriate knowledge about health and health behaviors from the stage of primary education.

One systematic review and metanalysis on health behavior patterns reported no sex differences in health risk behaviors [90], while others demonstrated that men are likely to engage in risky health behaviors [28, 32, 83]. Moreover, scholars reported that men have low levels of health literacy than those of women and may be more likely to engage in inadequate health behaviors [91]. The findings of the present study confirmed the past evidence on this topic.

In addition, the current results indicated that participants under Cluster 1 expressed high levels of happiness, which is in line with those of previous studies in other countries [23–25, 92]. The WHO has described a holistic approach to health through interventions for modifiable combinations of health behaviors. Furthermore, many countries have demonstrated that certain health behavior patterns exert synergistic effects on mortality, morbidity, subjective well-being, and happiness. As such, many of these countries have exerted effort to adopt integrated strategies for promoting health behavior patterns [25, 34–36, 54, 55, 92]. Nevertheless, researchers have yet to reach a consensus on the direction of the relationship between happiness and health behaviors. A few studies support the notion that levels of happiness predict health behaviors [93, 94], whereas others argue that the relationship is bidirectional [4, 95] and

that the direction of causality between happiness and health behaviors differs according to individual attributes [24, 96].

The current study is cross-sectional in nature. As such, it was unable to examine the causal relationship between health behaviors and happiness. Although engaging in health behaviors may generate positive emotions and increase happiness, a possibility exists that people engage in health habits, because they are happy and have a positive psychology. Therefore, further research is required to elucidate the underlying mechanisms of the relationship between health behavior and happiness. Nevertheless, the current study confirmed the relationship between high levels of happiness and health behaviors and demonstrated that health risk behaviors were more prevalent among those who are employed or with low levels of SES. If health behaviors lead to high levels of well-being, as indicated in the previous literature [23, 37], then endeavoring to assist the structuring of the private and work lives of employed Japanese individuals in a manner that enables employees to adopt health behaviors may be important for stakeholders. For this goal, policy interventions may be required.

Meanwhile, this study found that 33% of the participants had depressive tendencies. In a recent study, approximately 25% of Japanese adults suffer from mental health problems, such as anxiety disorders and mood disorders [97]. Therefore, although the prevalence of depression in the samples of the present study was slightly higher than that for mental health problems in the Japanese population, it did not deviate significantly from the trend.

## Limitations

This study has its limitations. First, analyses were based on cross-sectional data; thus, further investigation using longitudinal data is required to establish the causal relationships between happiness and health behaviors.

Second, the valid response rate was relatively low, because data from a single wave only (i.e., 2017) from a long-term panel dataset was used. The reality is that the youngest participants at baseline have gotten older. In addition, a certain percentage of the participants dropped out; thus, the possibility of sample attrition bias cannot be ruled out. Moreover, differences in age and sex distribution were observed between the included participants and those with missing data. Therefore, caution should be exercised when interpreting the findings.

Third, this study was based on a secondary data analysis, that is, the constraints imposed by the selected study design could not be avoided, especially regarding the measurement tools used in the questionnaire survey. A few of the problems noted are the unclear timeframes for the measurement of alcohol consumption; items about physical activity that did not comprehensively inquire about the amount and intensity of exercise; or items for vegetables and fruit intake that did not comprehensively inquire about the specific amounts consumed. Moreover, the original surveys from which the data were extracted assessed the happiness construct through a subjective, single-item scale that relied solely on a self-reported response. Thus, this variable lacked a comprehensive exploration, which requires future research to use a multidimensional objective instrument that yields data that enable the development of effective and successful strategies for enhancing happiness. In addition, the data on health behaviors are reliant on self-reported information, instead of observational measures; thus, a social disability bias may exist in the responses.

Despite these limitations, prior research highlighted the lack of studies on a combination of habitual health-related behaviors [98, 99]. To fill this research gap, the present study revealed clusters of health behaviors among Japanese adults and established the association between these clusters and happiness. Based on the results, the final remark of the study points to the

need to develop intervention strategies that comprehensively target multiple health behaviors, which may aid in enhancing happiness among the general Japanese population.

## Conclusion

In summary, the current study identified the clusters of health behavior and socio-demographics associated with these clusters in the general Japanese population. Thus, belonging to the cluster of low-risk health behaviors (low fruit intake, low exercise) may be significantly linked to high levels of happiness. Based on the results, the synergistic effect of improving health behaviors can be efficiently induced by constructing measures with a compound and chain effect, instead of strategies that improve each health behavior.

## Acknowledgments

We are deeply grateful to the Keio University Panel Data Research Center for providing the JHPS and KHPS data.

## Author Contributions

**Conceptualization:** Miho Satoh.

**Formal analysis:** Miho Satoh.

**Investigation:** Miho Satoh.

**Methodology:** Miho Satoh.

**Project administration:** Miho Satoh.

**Writing – original draft:** Miho Satoh.

**Writing – review & editing:** Miho Satoh.

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
