## [Decision Letter · Decision Letter 0]

17 Aug 2021

PONE-D-21-10430

Clustering of health behaviors among Japanese adults and their association with socio-demographics and happiness

PLOS ONE

Dear Dr. Satoh,

Thank you for submitting your manuscript to PLOS ONE. After careful consideration, we feel that it has merit but does not fully meet PLOS ONE’s publication criteria as it currently stands. Therefore, we invite you to submit a revised version of the manuscript that addresses the points raised during the review process.

The manuscript has been evaluated by two reviewers, and their comments are available below.

In addition to concerns regarding the need for greater clarity in the reporting of the results, both reviewers raise concerns regarding the Discussion section. Specifically, the reviewers note the need for greater depth of discussion regarding the implications of the study while avoiding overstating findings for a cross-sectional study.

Could you please carefully revise the manuscript to address all comments raised?

We look forward to receiving your revised manuscript.

Kind regards,

Avanti Dey, PhD

Staff Editor

PLOS ONE

Journal Requirements:

Reviewers' comments:

Reviewer's Responses to Questions

**Comments to the Author**

1. Is the manuscript technically sound, and do the data support the conclusions?

Reviewer #1: Partly

Reviewer #2: Yes

2. Has the statistical analysis been performed appropriately and rigorously? 

Reviewer #1: Yes

Reviewer #2: Yes

3. Have the authors made all data underlying the findings in their manuscript fully available?

Reviewer #1: Yes

Reviewer #2: Yes

4. Is the manuscript presented in an intelligible fashion and written in standard English?

Reviewer #1: Yes

Reviewer #2: Yes

5. Review Comments to the Author

Reviewer #1: This study is a secondary analysis of cross-sectional data from 1,554 questionnaires (participants aged 27-65) collected in the Japanese Household Panel Survey and the Keio Household Panel Survey. The study aims were to identify clustering of healthy behavior patterns, the association between socio-demographic characteristics and these clusters, and the association among clusters, number of health behaviors, and happiness among Japanese adults. The investigator used latent class analysis to identify the clusters, latent regression to investigate associations with socio-demographic characteristics, and logistic regression to examine the relationship between happiness and the clusters, controlling for demographic characteristics. Two health behavior clusters were identified: “moderately healthy” (54.0%) and “unhealthy (high alcohol, poor nutrition, 46.0%). The “moderately healthy” cluster was significantly related with higher odds of happiness (OR=1.425, CI=1.146−1.772, p=0.001). Participants who engaged in 4, 5, and 6 healthy behaviors had about three times higher odds of happiness (OR=3.239, CI=1.398−7.505, p=0.006; OR=3.886, CI=1.621−9.312, p=0.002; and OR=2.968, CI=1.057−8.332, p=0.039, respectively) compared to those who engaged in less healthy behaviors. This study strengthens the literature on the clustering of health behaviors and additionally describes the identified clusters in association with happiness and demographic factors.

Lines 62-64 – Confusing, please re-write

Lines 65-68 this paragraph suggests that health behaviors, both + and -, may co-occur, but the evidence cited is for clusters of unhealthy behaviors. Please rewrite

references 3-34 are the only two that dichotomized lower vs high risk lifestyle behaviors and inspected the association with well-being, which is not setting the bar too high.

Lines 72-74 You need a reference backing up your statement

Line 135-138 – explain how age and SES were entered/defined. Was age entered as a continuous variable? All your other variables are dichotomized, why not age? Your SES break-down has no variability (High 94%), this is problematic, could you break down at a different point? Please explain the rationale.

After Line 188 – Table 2

Your labels are wrong from Vegetable intake to Physical activity – your % add to 100%, but your descriptors suggest otherwise (e.g should be Every day vs. 6 or less days a week)

Lines 207-210 Age was a significant predictor of cluster 1, but you do not mention it in the results or the discussion.

Line 316 remove the word “mainly”

In limitations discuss your low response rate (valid response rate: 31.1%) and the high-income level of your sample.

Line 222-227 – In your discussion do not overstate your findings, you cannot prove causality nor directionality. Expand the implications of using cross-sectional data ONLY.

Acknowledge the efforts to evaluate directionality and its implications. The few studies that have examined the directionality within the happiness-health relationship appear to support the existence of the Top-Down Effect that suggests that happiness (a predisposition to color your life experiences) may directly impact health. However, studies that have provided evidence for the Top-Down Effect (Happiness may be impacted by our life experiences) have had test populations of primarily young adults (Kong, Hu, et al., 2015; Ngamaba et al., 2017; Schwerdtfeger et al., 2017). In contrast, the few studies that have examined the relationship in older adults (over the retirement age) provide some evidence of a Bottom-Up Effect (Gana et al., 2013 and from your reference list Moriyama et al). Several researchers suggests there is support for a bidirectional model (Feist, Bodner, Jacobs, Miles, & Tan, 1995; Diener, Lucas, & Scollon, 2006; Schimmack, Schupp, & Wagner, 2008).

Suggested references to add

1. Feist, G. J., Bodner, T. E., Jacobs, J. F., Miles, M., & Tan, V. (1995). Integrating Top-Down and Bottom-Up Structural Models of Subjective Well-Being : A Longitudinal Investigation. Journal of Personality and Social Psychology, 68(1), 138–150. doi: 10.1037/0022-3514.68.1.138

2. Gana, K., Bailly, N., Saada, Y., Joulain, M., Trouillet, R., Hervé, C., & Alaphilippe, D. (2013). Relationship between life satisfaction and physical health in older adults: a longitudinal test of cross-lagged and simultaneous effects. Health Psychology, 32(8), 896.

3. Leonardi, F., Spazzafumo, L., & Marcellini, F. (2005). Subjective well-being: The constructionist point of view. A longitudinal study to verify the predictive power of top-down effects and bottom-up processes. Social Indicators Research, 70(1), 53–77. doi: 10.1007/s11205-005-5016-7

4. Ngamaba, K. H., Panagioti, M., & Armitage, C. J. (2017). How strongly related are health status and subjective well-being? Systematic review and meta-analysis. European Journal of Public Health, 27(5), 879–885. doi: 10.1093/eurpub/ckx081

5. DIENER, E, LUCAS, R E, & SCOLLON, Christie N. (2006). Beyond the hedonic treadmill: Revising the adaptation theory of well-being. American Psychologist, 61(4), 305-314. https://ink.library.smu.edu.sg/soss_research/9 21

6. Schimmack, U., Schupp, J., & Wagner, G. G. (2008). The influence of environment and personality on the affective and cognitive component of subjective well-being. Social indicators research, 89(1), 41-60.

7. Røysamb, E., Tambs, K., Reichborn-Kjennerud, T., Neale, M. C., & Harris, J. R. (2003). Happiness and health: Environmental and genetic contributions to the relationship between subjective well-being, perceived health, and somatic illness. Journal of Personality and Social Psychology, 85(6), 1136–1146. doi: 10.1037/0022-3514.85.6.1136

8. Koivumaa-Honkanen, H., Honkanen, R., Viinamäki, H., Heikkilä, K., Kaprio, J., & Koskenvuo, M. (2000). Self-reported life satisfaction and 20-year mortality in healthy finnish adults. American Journal of Epedimiology, 152(10), 983–991.

9. Koivumaa-Honkanen, H., Kaprio, J., Honkanen, R. J., Viinamäki, H., & Koskenvuo, M. (2005). The stability of life satisfaction in a 15-year follow-up of adult Finns healthy at baseline. BMC Psychiatry, 5, 1–8. doi: 10.1186/1471-244X-5-4

10. Lyubomirsky, S., King, L., & Diener, E. (2005). The benefits of frequent positive affect: Does happiness lead to success? Psychological Bulletin, 131(6), 803–855. doi: 10.1037/0033-2909.131.6.803

Reviewer #2: This paper looks at the potential clustering of health behaviours and the association of these clusters with sociodemographic variables and happiness in Japanese adults, using data from two household surveys.

I think this is an interesting topic and paper, however I have some issues with both the measures and dichotomization of health behaviours along with the labelling of cluster 1. I also think the authors could go further in depth in both the introduction and discussion in order to strengthen the paper. My specific comments on the paper are outlined below:

Abstract

1. It would be helpful to split the abstract into subsections (intro, method, results, discussion) to make it easier to read.

2. ‘In total, 86.4% participants showed health behaviours simultaneously’ – I think this phrase needs rewording as it is not clear what is meant by it.

3. Are the confidence intervals 95% CIs? This needs stating.

4. You mention that participants who engaged in more than four behaviours and then report 3 Odds ratios – what do these relate to?

5. The conclusion seems to state that the associations are causal – by increasing health behaviours, then people will be happier. But this could also be the other way around, where happier people are more likely to perform health behaviours

Introduction

6. It would be useful to define happiness as it has been used in other studies – your introduction seems to state that it is a sub factor of wider wellbeing, but it would be useful to understand how it is being operationalized in the paper.

7. You could go further when describing how health behaviours may be interrelated, at the moment it feels like the introduction as a whole is lacking in some depth and could go further in the justification of why the study is important and why you have focused on these specific variables to look at.

8. Why focus on happiness? Why not use general wellbeing or a negative psychological variable such as distress? You mention that some other studies have looked at happiness, but this seems to be a very broad factor to focus on and I think you need to go further with why you decided to focus on this.

9. You refer to ‘human capital’ but don’t explain what this is – I am unfamiliar with this term and so it would be useful to explain what you mean by this and why it is important.

Method

10. The data used in this paper is from 2017 – is this the most recent data from these household surveys?

11. The questionnaire refers to ‘recent’ alcohol consumption – were participants given a more specific timeframe than this (i.e. In the past month or six months)? If not, then participants timeframes when answering this question may have differed from each other.

12. Were participants given a comparison amount to understand the ml of alcohol or were they asked about the type of alcohol? – 180ml of a spirit is a very different amount compared to wine, or does this refer to the alcoholic content of the drink? More detail is needed here. Also, is the 23gm of alcohol a standard measure of alcohol consumption? Most of the alcohol research I am familiar with uses units rather than grams in this way.

13. Participants were not questioned about the amount of cigarettes so you cannot differentiate between heavy vs. light smokers which is a limitation. As a whole, I think because you are using a secondary source of data, you are limited to the questions that were asked and this needs to be considered in the discussion.

14. Is the dichotomy for fruit and veg consumption none vs. any? This needs to be stated more clearly in the methods.

15. Again, there doesn’t seem to be a question on how much exercise participants did on the days they exercised so you cannot differentiate a ten-minute workout to one lasting an hour. – this is a limitation of the questions used in the survey.

16. Happiness is a single-item measure that was split using a median split – this needs to be mentioned in the limitations section. Were participants given any reference information to better understand what was meant by happiness in the study?

17. On line 156, p8 you mention depression but this has not been mentioned previously.

Results

18. The total participant number is reduced down to 1554, presumably due to missing or invalid data – where was the data missing which meant that you could not use all participants in the analyses? Is this sample still representative once you account for the missing data? – The original household survey age range was 20-69, however in your sample they are now 27-65 so I wonder if younger people are still represented in the analyses.

19. Only 6% of participants report low SES – I think this is important as we know that SES is associated with health behaviours. Does this indicate that the sample may not be representative of the whole Japanese population?

20. 44% of participants report depression – this seems very high to me, what was the measure of depression used? I think it would be worth discussing whether you expected nearly half of the sample to report depression in the paper.

21. It would be useful to report the Ns in the text when discussing the health behaviour prevalence information.

22. I think it would be useful to flip around the physical activity percentage as all the other percentages you report relate to healthy rather than unhealthy behaviours – 18% of participants reported physical activity on more than 2 days per week.

23. Why are fruit and vegetable consumption reported separately and not combined into one measure?

24. I’m not sure about the labelling of Cluster 1 as ‘moderately healthy’ – the label you use for cluster 2 is more descriptive whereas ‘moderately healthy’ seems to be more of a judgement. Looking at the behaviours reported, I’m not sure I would classify these individuals as ‘moderately’ healthy. In my opinion, it would be better to use a more descriptive label for cluster 1.

25. P11, line 197, ‘high level of inadequate alcohol consumption’ – you refer to ‘inadequate alcohol consumption’ in a few places and I think this needs to be reworded to ‘higher level of alcohol consumption’ or similar.

26. What is the comparison group used for employment status?

27. Lines 216-217, p12, need to be clear that these results relate to 4, 5 and 6 health behaviours. Currently this is unclear.

Discussion

28. I think it is worth pointing out that while cluster 1 refers to ‘healthier’ behaviours than cluster 2, neither group is particularly healthy and participants in the study tended to report fairly low levels of health behaviours.

29. It would be worth discussing how representative the sample is of the Japanese population when you consider the participants whose data were analysed – the youngest participant was 27 and the vast majority of participants report high levels of SES, so I wonder if the results have issues with self-sampling bias.

30. You mention that stress = poorer health and that this is likely to be related to both health behaviours and happiness. Do you think that by measuring happiness, this is a kind of proxy measure where you are actually assessing stress levels?

You also mention that by increasing health behaviours, this may increase happiness. But I wonder if the issue is actually that people are too stressed to carry out health behaviours and this reduces happiness, therefore wouldn’t reducing stress be the key factor to target in order to have the biggest impact over health and wellbeing?

31. As mentioned in some of my earlier comments, I think you need to go further when discussing the limitations of the study including the single item measure of happiness, the missing data, and the issues around using secondary sources of data.

6. PLOS authors have the option to publish the peer review history of their article (what does this mean?). If published, this will include your full peer review and any attached files.

Reviewer #1: No

Reviewer #2: No

---

## [Author Response · Author response to Decision Letter 0]

13 Sep 2021

Reviewer #1

Lines 62-64 – Confusing, please re-write

I appreciate your comment. The Introduction section has been generally revised to make its arguments and content clearer. Lines 62-64 of the original draft were revised as follows [Lines 64-67].

Despite the plethora of information on happiness and the growing number of studies across the globe finding that greater engagement in a variety of desirable health behaviors is linked to higher happiness, few studies have examined the associations between health behaviors and happiness in the Japanese population [26, 27]. 

-------------

Lines 65-68 this paragraph suggests that health behaviors, both + and -, may co-occur, but the evidence cited is for clusters of unhealthy behaviors. Please rewrite

Thank you for bringing this to my attention. Lines 65-68 of the original draft were revised as follows [Lines 70-74].

Indeed, some research in Europe and the United States of America have found that multiple healthy behaviors tend to occur interrelatedly and that the degree of health consciousness has a cascading effect on all types of health behaviors [28-30]. In other words, one healthy behavior can trigger a chain reaction of other healthy behaviors, and the contrary is also true.

---------------

references 3-34 are the only two that dichotomized lower vs high risk lifestyle behaviors and inspected the association with well-being, which is not setting the bar too high.

Thank you very much for this commentary. I reviewed these references and corrected their citations; specifically, the reference numbers were changed: 33 to 36, 34 to 37 [Lines 77-81].

Multiple health behaviors have synergistic effects, and therefore research on the clustering of multiple health behavior patterns and their relationship to health outcomes is increasing [28, 30, 32-36]. Further, studies have analyzed the synergistic effects of multiple health behaviors on happiness [23, 37], but findings on these effects remain scarce and they have not been examined in Japan.

--------

Lines 72-74 You need a reference backing up your statement

Thank you for this remark. Lines 72-74 in the first draft were revised as follows [Lines 59-63].

In health research, there have also been numerous studies identifying factors associated with happiness, including morbidity and mortality [4, 14], absence of disease [4], and subjective health and psychological health [15]. Various health behaviors have also been found to be associated with happiness, such as being physically active [16, 17], consuming vegetables and fruits [18-20], not smoking [21, 22], and consuming alcohol in moderation [23-25].

-------------

Line 135-138 – explain how age and SES were entered/defined. Was age entered as a continuous variable? All your other variables are dichotomized, why not age? Your SES break-down has no variability (High 94%), this is problematic, could you break down at a different point? Please explain the rationale.

Thank you for bringing this to my attention. Since I had entered age into the model as a continuous variable without sufficient consideration, I categorized and reanalyzed the age distribution. I have also reviewed the distribution of SES, the SES categories, and recreated categories based on quartiles [Lines 169-173].

Based on age, the participants were divided into three groups: younger (27-44), middle (45-55) and older (56-65). Regarding SES, participants were asked to describe the disposable income per household, so the responses were divided into three groups based on the quartile: lower (<2,150 K, where K is 1,000 yen), middle (2,150 K to<4,040 K) and higher (≥4,040 K).

------------------

After Line 188 – Table 2

Your labels are wrong from Vegetable intake to Physical activity – your % add to 100%, but your descriptors suggest otherwise (e.g should be Every day vs. 6 or less days a week)

Thank you very much for this commentary. Table 2 indicated the raw number and the percentages for participants with higher risk behavior or lower risk behavior. I have now revised the labels in the Table to ensure greater clarity for the classification of higher risk behavior and lower risk behavior by category.

---------------

Lines 207-210 Age was a significant predictor of cluster 1, but you do not mention it in the results or the discussion.

Thank you for your commentary. Although I did discuss about the relationship between age and the Clusters, my discussion about this topic was insufficient and was unclear. Thus, I partly revised Lines 207-210 in the original draft as follows [Lines 284-295].

Indeed, as people get older, they develop more health awareness and attempt to engage in healthier behaviors and lifestyles [32,61]. Among the Japanese people, those older tend to have more health awareness and adopt healthier lifestyles compared with younger people [41,62]. Still, other research revealed that people in their 30s to 50s are less likely to be aware of their own health because of increasing roles and responsibilities in both family and work life [41,63], while younger people have lower awareness since they do not have subjective symptoms of chronic or other diseases [64,65]. Moreover, studies showed that to reduce the risk of non-communicable diseases (e.g., diabetes, hypertension, or cardiovascular diseases) it is important to take measures aimed at changing multiple health behaviors [66,67]. Therefore, since health-promoting behaviors may have a significant impact on the future health status of young people, it may be necessary to develop and promote health awareness-related initiatives among the younger groups of the Japanese population that encourage their engagement in health-promoting behaviors.

---------------

Line 316 remove the word “mainly”

Thank you for your advice. I have followed your recommendation and removed ‘mainly’ from this sentence.

------------------

In limitations discuss your low response rate (valid response rate: 31.1%) and the high-income level of your sample.

I appreciate your comment and advice. I added the following description about the low response rate in my study and the aforementioned sample bias to Lines 359-363. Regarding SES, I have reconfirmed and the distribution and category division, recreating them based on quartiles. Thus, I saw no need for including more information regarding the topic of SES in the Limitations section [Lines 361-365].

Second, the valid response rate was relatively low in this study; since I used data only from a single wave (i.e., 2017) of a long-term panel dataset, it is a reality that the youngest participants at baseline are getting older. Moreover, a certain percentage of participants dropped out, and I cannot rule out the possibility of sample attrition bias. Therefore, caution should exerted when interpreting the study findings.

------------------

Line 222-227 – In your discussion do not overstate your findings, you cannot prove causality nor directionality. Expand the implications of using cross-sectional data ONLY.

Acknowledge the efforts to evaluate directionality and its implications. The few studies that have examined the directionality within the happiness-health relationship appear to support the existence of the Top-Down Effect that suggests that happiness (a predisposition to color your life experiences) may directly impact health. However, studies that have provided evidence for the Top-Down Effect (Happiness may be impacted by our life experiences) have had test populations of primarily young adults (Kong, Hu, et al., 2015; Ngamaba et al., 2017; Schwerdtfeger et al., 2017). In contrast, the few studies that have examined the relationship in older adults (over the retirement age) provide some evidence of a Bottom-Up Effect (Gana et al., 2013 and from your reference list Moriyama et al). Several researchers suggests there is support for a bidirectional model (Feist, Bodner, Jacobs, Miles, & Tan, 1995; Diener, Lucas, & Scollon, 2006; Schimmack, Schupp, & Wagner, 2008).

I appreciate your commentary and advice. I agreed with your appointments, so I have revised the Discussion and the Limitations section as follows [Lines 344-355] and [Lines 358-360].

As the current study was cross-sectional, the causal relationship between health behaviors and happiness could not be examined. However, the current study confirmed the relationship between high levels of happiness and healthy behaviors and showed that health risk behaviors were more prevalent among those who had jobs or had lower SES. If healthy behaviors lead to higher levels of well-being as indicated in previous research [23, 37], it may be important for stakeholders to endeavor to assist the structuring of the private and work lives of employed Japanese individuals in a way that enables employees to adopt these healthy behaviors, a goal for which policy interventions may be required.

Meanwhile, although engaging in health behaviors may generate positive emotions and increase happiness, it is also possible that people engage in health habits because they are happy and have a positive psychology. Therefore, further research is needed to elucidate the underlying mechanism of the relationship between health behavior and happiness.

Limitations

Regarding study limitations, first, the analyses were based on cross-sectional data; thus, further investigation using longitudinal data is needed to understand the causal relationships between happiness and health behaviors.

Suggested references to add

I greatly appreciate your advice. I reviewed the References section as per your suggestion and reviewed other relevant literature. This resulted in my study now citing some of the references you have provided and some additional studies I have reviewed.

1. Feist, G. J., Bodner, T. E., Jacobs, J. F., Miles, M., & Tan, V. (1995). Integrating Top-Down and Bottom-Up Structural Models of Subjective Well-Being: A Longitudinal Investigation. Journal of Personality and Social Psychology, 68(1), 138–150. doi: 10.1037/0022-3514.68.1.138

2. Gana, K., Bailly, N., Saada, Y., Joulain, M., Trouillet, R., Hervé, C., & Alaphilippe, D. (2013). Relationship between life satisfaction and physical health in older adults: a longitudinal test of cross-lagged and simultaneous effects. Health Psychology, 32(8), 896.

3. Leonardi, F., Spazzafumo, L., & Marcellini, F. (2005). Subjective well-being: The constructionist point of view. A longitudinal study to verify the predictive power of top-down effects and bottom-up processes. Social Indicators Research, 70(1), 53–77. doi: 10.1007/s11205-005-5016-7

4. Ngamaba, K. H., Panagioti, M., & Armitage, C. J. (2017). How strongly related are health status and subjective well-being? Systematic review and meta-analysis. European Journal of Public Health, 27(5), 879–885. doi: 10.1093/eurpub/ckx081

5. DIENER, E, LUCAS, R E, & SCOLLON, Christie N. (2006). Beyond the hedonic treadmill: Revising the adaptation theory of well-being. American Psychologist, 61(4), 305-314. https://ink.library.smu.edu.sg/soss_research/9 21

6. Schimmack, U., Schupp, J., & Wagner, G. G. (2008). The influence of environment and personality on the affective and cognitive component of subjective well-being. Social indicators research, 89(1), 41-60.

7. Røysamb, E., Tambs, K., Reichborn-Kjennerud, T., Neale, M. C., & Harris, J. R. (2003). Happiness and health: Environmental and genetic contributions to the relationship between subjective well-being, perceived health, and somatic illness. Journal of Personality and Social Psychology, 85(6), 1136–1146. doi: 10.1037/0022-3514.85.6.1136

8. Koivumaa-Honkanen, H., Honkanen, R., Viinamäki, H., Heikkilä, K., Kaprio, J., & Koskenvuo, M. (2000). Self-reported life satisfaction and 20-year mortality in healthy finnish adults. American Journal of Epedimiology, 152(10), 983–991.

9. Koivumaa-Honkanen, H., Kaprio, J., Honkanen, R. J., Viinamäki, H., & Koskenvuo, M. (2005). The stability of life satisfaction in a 15-year follow-up of adult Finns healthy at baseline. BMC Psychiatry, 5, 1–8. doi: 10.1186/1471-244X-5-4

10. Lyubomirsky, S., King, L., & Diener, E. (2005). The benefits of frequent positive affect: Does happiness lead to success? Psychological Bulletin, 131(6), 803–855. doi: 10.1037/0033-2909.131.6.803

 

Reviewer #2

Abstract

1. It would be helpful to split the abstract into subsections (intro, method, results, discussion) to make it easier to read.

I appreciate your advice. Since I concur with you, I have added subsections to the Abstract.

2. ‘In total, 86.4% participants showed health behaviours simultaneously’ – I think this phrase needs rewording as it is not clear what is meant by it.

I appreciate your comment. After due consideration, I deemed that including a discussion on the relationship between the number of unhealthy behaviors and well-being in the current paper would make the argument of the manuscript somewhat confusing; thus, I decided not to include this topic in the study. However, I overlooked a portion of the content related to this topic that needed to be excluded; I have now removed this sentence, and I apologize for this mistake.

3. Are the confidence intervals 95% CIs? This needs stating.

Thanks for this remark. I revised “CI” to “95% CIs” throughout the whole paper.

4. You mention that participants who engaged in more than four behaviours and then report 3 Odds ratios – what do these relate to?

Thank you very much for this question. After due consideration, I deemed that including a discussion on the relationship between the number of unhealthy behaviors and well-being in the current paper would make the argument of the manuscript somewhat confusing; thus, I decided not to include this topic in the study. However, I overlooked a portion of the content related to this topic that needed to be excluded; I have now removed this sentence, and I apologize for this mistake.

5. The conclusion seems to state that the associations are causal – by increasing health behaviours, then people will be happier. But this could also be the other way around, where happier people are more likely to perform health behaviours

I would like to show my appreciation for your advice. Since I agree with you, I have revised the Conclusion of the abstract to concur with your appointments, as follows.

Conclusion: This study identified the clusters of health behavior patterns among Japanese adults, and an association between the lower health risk cluster and higher happiness was indicated. However, the causality of the relationship between health behavior and happiness was not clarified, highlighting the need for further research to elucidate its underlying mechanisms.

Introduction

6. It would be useful to define happiness as it has been used in other studies – your introduction seems to state that it is a sub factor of wider wellbeing, but it would be useful to understand how it is being operationalized in the paper.

Thank you very much for this commentary. I now added a definition of happiness, as follows [Lines 50-54]. 

Conceptually speaking, happiness refers to one’s overall appreciation of life as a whole and to a subjective state of mind [9], and has often been deemed as an ultimate life goal [5]. According to Diener [10], happiness is synonymous with subjective well-being and can be regarded as a global term for the various types of evaluations that people make, both positive and negative, about their lives.

7. You could go further when describing how health behaviours may be interrelated, at the moment it feels like the introduction as a whole is lacking in some depth and could go further in the justification of why the study is important and why you have focused on these specific variables to look at.

Thanks for this suggestion. I have generally modified the Introduction section based on your commentaries as well as those from Review 1. The part you pointed out was revised and added in the Introduction [Lines 68-81] and Variables section [Lines 110-114] as follows.

Introduction:

In addition, many studies have investigated the relationship between individual health behaviors and physical or psychological health status in isolation; nonetheless, I know that lifestyles are made up of multiple health behaviors. Indeed, some research in Europe and the United States of America have found that multiple healthy behaviors tend to occur interrelatedly and that the degree of health consciousness has a cascading effect on all types of health behaviors [28-30]. In other words, one healthy behavior can trigger a chain reaction of other healthy behaviors, and the contrary is also true. Moreover, the simultaneous occurrence of multiple health behaviors was attributed to various mechanisms, including the commonality of multiple factors associated with the behaviors, and the fact that one behavior may be a coping strategy for another [31].

Multiple health behaviors have synergistic effects, and therefore research on the clustering of multiple health behavior patterns and their relationship to health outcomes is increasing [28, 30, 32-36]. Further, studies have analyzed the synergistic effects of multiple health behaviors on happiness [23, 37], but findings on these effects remain scarce and they have not been examined in Japan.

Variables:

The health behaviors that have been emphasized in previous studies are smoking, alcohol consumption, diet, and exercise [28-30, 38-40], with more recent ones having focused on the relationship between vegetable and fruit intake and physical and psychological health [18-20, 28-30]. In addition, studies have analyzed whether or not people have the habit of eating breakfast and how this associates with one’s physical and psychological health [28, 29].

8. Why focus on happiness? Why not use general wellbeing or a negative psychological variable such as distress? You mention that some other studies have looked at happiness, but this seems to be a very broad factor to focus on and I think you need to go further with why you decided to focus on this.

I sincerely oblige your questionings. I have generally modified the Introduction section based on your commentaries as well as those from Review 1, modifying the initial paragraph, Lines 48-56, to be as follows. I hope it has given space for the questions you have given rise to above to be discussed.

Worldwide, the term “happiness,” which has been considered as a key element of individuals’ well-being, productivity, and quality of life, has received much attention [1-8]. Conceptually speaking, happiness refers to one’s overall appreciation of life as a whole and to a subjective state of mind [9], and has often been deemed as an ultimate life goal [5]. According to Diener [10], happiness is synonymous with subjective well-being and can be regarded as a global term for the various types of evaluations that people make, both positive and negative, about their lives. In general, happiness is a concept related to well-being that can be comprehensively evaluated regarding its positive and negative aspects; thus, it may serve as an effective indicator for measuring individuals’ overall satisfaction with their situation in life.

9. You refer to ‘human capital’ but don’t explain what this is – I am unfamiliar with this term and so it would be useful to explain what you mean by this and why it is important.

Thanks for this bringing this to my attention. I have generally modified the Introduction section based on your commentaries as well as those from Review 1; during the process, the term “human capital” was removed.

Method

10. The data used in this paper is from 2017 – is this the most recent data from these household surveys?

Thanks for this question. Yes, the 2017 dataset is the most recent data available at the time I wrote this manuscript. However, the 2019 dataset is ready for use. So, I will confirm the replication of this study in the future. 

11. The questionnaire refers to ‘recent’ alcohol consumption – were participants given a more specific timeframe than this (i.e. In the past month or six months)? If not, then participants timeframes when answering this question may have differed from each other.

Thanks for highlighting this topic. The survey data did not indicate a specific timeframe for the alcohol consumption measurement; thus, I have added the following information to the Limitations section [Lines 366-371].

Third, this study was based on a secondary data analysis; namely, I could not escape the constraints imposed by the chosen study design, especially regarding the measurement tools used by the questionnaire survey. Some of the problems that they have include unclear timeframes for measuring alcohol consumption, having items about physical activity that did not comprehensively inquire about exercise amount and intensity, or about vegetables and fruit intake that did not comprehensively inquire about the specific amounts that participants ate.

12. Were participants given a comparison amount to understand the ml of alcohol or were they asked about the type of alcohol? – 180ml of a spirit is a very different amount compared to wine, or does this refer to the alcoholic content of the drink? More detail is needed here. Also, is the 23gm of alcohol a standard measure of alcohol consumption? Most of the alcohol research I am familiar with uses units rather than grams in this way.

Thank you for your comment. Respondents were asked to convert their answers regarding drinking to the amount of sake that this would represent; namely, their drinking quantities of other types of alcoholic beverages were to be converted to what they would represent based on a sake bottle. This allowed for calculating the alcohol content that they consumed. To further describe this in the text, I have added the follow description to the Variables section [Lines119-126].

Alcohol consumption was assessed using two questions. The first question was “How was your recent alcohol consumption habit?” Which had a scale with the following responses: never, a few times a month, 1−2 times a week, more than three times a week, and every day; the second question was “What is your average alcohol consumption in day when converted to sake?” Which had a scale with the following responses: less than 180 ml, 180−360 ml, 360−540 ml, 540−720 ml, 720−900 ml, and more than 900 ml. As sake contains 23 g of alcohol per 180 ml, responses were coded as 0 for “more than 23 g of alcohol daily” and 1 for “no-alcohol consumption or less than 23 g of alcohol daily.”

13. Participants were not questioned about the amount of cigarettes so you cannot differentiate between heavy vs. light smokers which is a limitation. As a whole, I think because you are using a secondary source of data, you are limited to the questions that were asked and this needs to be considered in the discussion.

Thanks for this commentary. In this survey, there seemed to be no consideration regarding the number of cigarettes smoked; however, I have referred to the WHO’s view, which describes that smoking even one cigarette per day can have a negative impact on health [Lines 127-131].

Smoking status was assessed using a single question: “Do you smoke cigarettes?” Which had a scale with the following responses: I smoke, I smoke sometimes, I have quit smoking, and never. The World Health Organization (WHO) recommends people to not smoke tobacco in order to maintain a good health [42, 43]; responses were coded as 1 for “non-smokers,” as those who were not current smokers, and 0 for “smokers,” as those who were current smokers. 

14. Is the dichotomy for fruit and veg consumption none vs. any? This needs to be stated more clearly in the Variables.

Thanks for this question. I have added the following sentence to Line 136-137 to ensure greater clarity: Responses were coded as 0 for “less than 2 times daily” and 1 for “more than 2 times daily.”

15. Again, there doesn’t seem to be a question on how much exercise participants did on the days they exercised so you cannot differentiate a ten-minute workout to one lasting an hour. – this is a limitation of the questions used in the survey.

Thanks for this description. The participants were not asked to provide responses about the amount and intensity of exercise in which they engaged in; thus, I have added the following description to the Limitation section [Lines 366-371].

Third, this study was based on a secondary data analysis; namely, I could not escape the constraints imposed by the chosen study design, especially regarding the measurement tools used by the questionnaire survey. Some of the problems that they have include unclear timeframes for measuring alcohol consumption, having items about physical activity that did not comprehensively inquire about exercise amount and intensity, or about vegetables and fruit intake that did not comprehensively inquire about the specific amounts that participants ate.

16. Happiness is a single-item measure that was split using a median split – this needs to be mentioned in the limitations section. Were participants given any reference information to better understand what was meant by happiness in the study?

Thank you for your comment. The analyzed surveys did not provide further information about the term “Happiness.” Therefore, I have added the following description to the Limitation section [Lines 371-376].

Moreover, the original surveys from which my data were extracted assessed the happiness construct through a subjective, single-item scale that relied solely on self-reported responses; thus, this variable was not comprehensively explored, making it necessary for future research to use a multi-dimensional objective instrument that yields data which may allow for the development of more effective and successful strategies.

17. On line 156, p8 you mention depression but this has not been mentioned previously.

Thank you very much for highlighting this gap, and I sincerely apologize for the lack of information on the topic. I added the following description to the Variables section [Lines 155-162].

Psychological health

As psychological health can influence health behaviors and happiness, I used participants’ depression level as a control variable. Psychological health was measured using the Japanese version of the 12-item General Health Questionnaire (GHQ) [45, 46], and each item was rated using a 4-point Likert scale. To assess respondents’ depression level, the score categories (1= ‘less than usual’, 2= ‘no more than usual’, 3= ‘rather more than usual’, and 4= ‘much more than usual’) were converted into binary values (0, 0, 1, and 1), where higher totals score indicated more severe psychological distress. The depression for cut-off scores between 3 and 4 was used [45, 46].

Results

18. The total participant number is reduced down to 1554, presumably due to missing or invalid data – where was the data missing which meant that you could not use all participants in the analyses? Is this sample still representative once you account for the missing data? – The original household survey age range was 20-69, however in your sample they are now 27-65 so I wonder if younger people are still represented in the analyses.

I appreciate your comment. Since participants with incomplete data were excluded, I analyzed the responses of 1,554 completed questionnaires (valid response rate: 31.1%; this is described in the results section). I have now added the following description about low response rate and sample bias to the Limitation section [Lines 361-365].

Second, the valid response rate was relatively low in this study; since I used data only from a single wave (i.e., 2017) of a long-term panel dataset, it is a reality that the youngest participants at baseline are getting older. Moreover, a certain percentage of participants dropped out, and I cannot rule out the possibility of sample attrition bias. Therefore, caution should exerted when interpreting the study findings.

19. Only 6% of participants report low SES – I think this is important as we know that SES is associated with health behaviours. Does this indicate that the sample may not be representative of the whole Japanese population?

Thank you for this question. Based on peer review comments, I have reconfirmed the SES distribution, reexamined its categorization, and recreated categories based on quartiles. This highlighted a significant association between higher SES and health behaviors. Accordingly, I added information about this topic in Lines 234-238 and in Lines 312-316.

Results:

Meanwhile, participants who were married (OR=1.521, 95% CIs=1.130−2.047, p=0.006), had higher education (OR=2.333, 95% CIs=1.842−2.955, p=0.000), and had higher SES (OR=1.483, 95% CIs=1.083−2.030, p=0.014) had higher odds of belonging to Cluster 1.

Discussion:

The literature has already thoroughly discussed the socioeconomic gradient in health outcomes and health behaviors [77]. Specifically, multiple healthy behaviors were more prevalent among those with higher SES [69,78], whereas those with lower SES showed a greater prevalence of multiple health risk behaviors [28,69]. The sample of the current study confirmed these associations between multiple health behavior and SES.

20. 44% of participants report depression – this seems very high to me, what was the measure of depression used? I think it would be worth discussing whether you expected nearly half of the sample to report depression in the paper.

I appreciate your suggestion. After reviewing the GHQ cut-off points, I observed that 33% of all respondents showed depression. Still, I used GHQ only as a control variable, mostly because I assumed that the mental health status could affect happiness and health behaviors. Thus, I provided a description at the beginning of the Discussion about the characteristics of depression among the study sample, as follows [Lines 247-251].

This study showed that 33% of the participants had depressive tendencies. In a recent study, around 25% of the Japanese adults were estimated to suffer from mental health problems, including anxiety disorders and mood disorders [49]. Therefore, although the prevalence of depression in the sample of the present study was slightly higher than that for mental health problems in the Japanese population, it did not deviate significantly from the trend.

21. It would be useful to report the Ns in the text when discussing the health behaviour prevalence information.

Thanks for this remark. To address your commentary, I have added the numbers related to Cluster 1 and Cluster 2, and a Table for health behavior prevalence in both clusters: “Cluster 1 (n=817; 52.3%)” and “Cluster 2 (n=737; 47.7%).” I have also included Fig. 1 with data on the topic.

22. I think it would be useful to flip around the physical activity percentage as all the other percentages you report relate to healthy rather than unhealthy behaviours – 18% of participants reported physical activity on more than 2 days per week.

Thank you for your advice. I revised this topic as follows [Lines 210-214].

The prevalence of health behaviors is presented in Table 2. In total, 76.9% of the participants were non-smokers, and 38.5% reported no alcohol consumption or <23 g of alcohol consumption a day. Further, 46.2% ate vegetables more than 2 times daily, 7.1 % ate fruits more than 2 times daily, 75.3% reported having the habit of eating breakfast every day, and 17.8% reported engaging in physical activity more than two times a week.

23. Why are fruit and vegetable consumption reported separately and not combined into one measure?

Thank you for your pointing this out and providing this question. Vegetable intake and fruit intake were considered separate variables because they are not necessarily correlated in terms of quantity, frequency, or intake patterns, and it has been suggested that they may vary independently.

Reference: Ministry of Health, Labour and Welfare. National Health and Nutrition Survey [Internet]. [cited 2021 Sept 2]. Available from: https://www.mhlw.go.jp/content/10900000/000687163.pdf (in Japanese).

24. I’m not sure about the labelling of Cluster 1 as ‘moderately healthy’ – the label you use for cluster 2 is more descriptive whereas ‘moderately healthy’ seems to be more of a judgement. Looking at the behaviours reported, I’m not sure I would classify these individuals as ‘moderately’ healthy. In my opinion, it would be better to use a more descriptive label for cluster 1.

I appreciate your suggestion and agree with you. I have, thus reconsidered the label for Cluster 1 based on the results, having revised the topic as follows [Lines 219-222].

Cluster 1 (n=817; 52.3%) was labelled “low risk, poor fruit and exercise,” with participants in it being characterized by the lowest probability for smoking, moderately low alcohol consumption, the highest vegetable intake, regular breakfast intake, as well as low fruit intake and physical activity.

25. P11, line 197, ‘high level of inadequate alcohol consumption’ – you refer to ‘inadequate alcohol consumption’ in a few places and I think this needs to be reworded to ‘higher level of alcohol consumption’ or similar.

Thank you for this remark. I have reworded “high level of inadequate alcohol consumption” to “high level of alcohol consumption.” [Line 269]

26. What is the comparison group used for employment status?

Thank you for your comment. As per your remark, I have added the comparison group, “non-employed,” in the Table.

27. Lines 216-217, p12, need to be clear that these results relate to 4, 5 and 6 health behaviours. Currently this is unclear.

I appreciate your comment. After due consideration, I deemed that including a discussion on the relationship between the number of unhealthy behaviors and well-being in the current paper would make the argument of the manuscript somewhat confusing; thus, I decided not to include this topic in the study. However, I overlooked a portion of the content related to this topic that needed to be excluded; I have now removed this sentence (Lines 216-217, p12 in the first submitted manuscript), and I apologize for this mistake. I have also deleted the relevant table.

Discussion

28. I think it is worth pointing out that while cluster 1 refers to ‘healthier’ behaviours than cluster 2, neither group is particularly healthy and participants in the study tended to report fairly low levels of health behaviours.

I appreciate your advice. Accordingly, I have added the following description to the Discussion section [Lines 255-257].

Based on the distribution of health behavior implementation, participants in Cluster 1 showed lower health risk behaviors than in Cluster 2, although those in Cluster 1 were not completely healthy.

29. It would be worth discussing how representative the sample is of the Japanese population when you consider the participants whose data were analysed – the youngest participant was 27 and the vast majority of participants report high levels of SES, so I wonder if the results have issues with self-sampling bias.

Thank you very much for this commentary. Based on peer review comments, I have reconfirmed the SES distribution, reexamined its categorization, and recreated categories based on quartiles. This led me to conclude that there was no need add a description regarding this topic in the Limitation section.

30. You mention that stress = poorer health and that this is likely to be related to both health behaviours and happiness. Do you think that by measuring happiness, this is a kind of proxy measure where you are actually assessing stress levels?

You also mention that by increasing health behaviours, this may increase happiness. But I wonder if the issue is actually that people are too stressed to carry out health behaviours and this reduces happiness, therefore wouldn’t reducing stress be the key factor to target in order to have the biggest impact over health and wellbeing?

I appreciate your comment and advice, and I strongly agree with you. I do not think that happiness is a proxy measure of stress level. Moreover, since my study could not clarify the causality between health behaviors and happiness, the description I provided was somewhat misleading. Thus, I have revised the Discussion section to concur with the discussions you have brought forth, as follows [Lines 320-328].

Nonetheless, it has also been reported that many employed people tend to have unhealthy lifestyles because they are too busy with work to pay attention to their own health habits [41,82,83]. Moreover, work-related stress might influence the engagement in health risk behaviors, such as alcohol consumption [84,85]. The unique cultural background of Japan may also lead employed individuals to very frequently go out for drinks to interact socially and professionally with bosses and colleagues after work [86,87]; hence, those employed in Japan may be more likely to adopt unhealthy behaviors, highlighting the potential need for comprehensively analyzing the work lives of Japanese workers in order to improve their healthy behavior patterns.

31. As mentioned in some of my earlier comments, I think you need to go further when discussing the limitations of the study including the single item measure of happiness, the missing data, and the issues around using secondary sources of data.

I would like to demonstrate my appreciation for this commentary. To ensure that I followed your guidance, I have revised the Limitation section and the final portion of the Discussion section to ensure that this limitation is more clearly described, as follows [Lines 361-376] and [Lines 344-351].

Limitations:

Second, the valid response rate was relatively low in this study; since I used data only from a single wave (i.e., 2017) of a long-term panel dataset, it is a reality that the youngest participants at baseline are getting older. Moreover, a certain percentage of participants dropped out, and I cannot rule out the possibility of sample attrition bias. Therefore, caution should be exerted when interpreting the study findings. 

Third, this study was based on a secondary data analysis; namely, I could not escape the constraints imposed by the chosen study design, especially regarding the measurement tools used by the questionnaire survey. Some of the problems that they have include unclear timeframes for measuring alcohol consumption, having items about physical activity that did not comprehensively inquire about exercise amount and intensity, or about vegetables and fruit intake that did not comprehensively inquire about the specific amounts that participants ate. Moreover, the original surveys from which my data were extracted assessed the happiness construct through a subjective, single-item scale that relied solely on self-reported responses; thus, this variable was not comprehensively explored, making it necessary for future research to use a multi-dimensional objective instrument that yields data which may allow for the development of more effective and successful strategies.

Discussion:

As the current study was cross-sectional, the causal relationship between health behaviors and happiness could not be examined. However, the current study confirmed the relationship between high levels of happiness and healthy behaviors and showed that health risk behaviors were more prevalent among those who had jobs or had lower SES. If healthy behaviors lead to higher levels of well-being as indicated in previous research [23, 37], it may be important for stakeholders to endeavor to assist the structuring of the private and work lives of employed Japanese individuals in a way that enables employees to adopt these healthy behaviors, a goal for which policy interventions may be required.

---

## [Decision Letter · Decision Letter 1]

8 Nov 2021

PONE-D-21-10430R1Clustering of health behaviors among Japanese adults and their association with socio-demographics and happinessPLOS ONE

Dear Dr. Miho Satoh,

Thank you for submitting your manuscript to PLOS ONE. After careful consideration, we feel that it has merit but does not fully meet PLOS ONE’s publication criteria as it currently stands. Therefore, we invite you to submit a revised version of the manuscript that addresses the points raised during the review process.

We look forward to receiving your revised manuscript.

Kind regards,

Akihiro Nishi, M.D., Dr.P.H.

Academic Editor

PLOS ONE

Additional Editor Comments:

The editor has agreed the two reviewers' recommendation - one for major revision and the other minor revision. They suggest constructive modifications. Please do.

Reviewers' comments:

Reviewer's Responses to Questions

**Comments to the Author**

1. If the authors have adequately addressed your comments raised in a previous round of review and you feel that this manuscript is now acceptable for publication, you may indicate that here to bypass the “Comments to the Author” section, enter your conflict of interest statement in the “Confidential to Editor” section, and submit your "Accept" recommendation.

Reviewer #1: (No Response)

Reviewer #2: (No Response)

2. Is the manuscript technically sound, and do the data support the conclusions?

Reviewer #1: Yes

Reviewer #2: Yes

3. Has the statistical analysis been performed appropriately and rigorously? 

Reviewer #1: Yes

Reviewer #2: Yes

4. Have the authors made all data underlying the findings in their manuscript fully available?

Reviewer #1: Yes

Reviewer #2: Yes

5. Is the manuscript presented in an intelligible fashion and written in standard English?

Reviewer #1: Yes

Reviewer #2: Yes

6. Review Comments to the Author

Reviewer #1: The manuscript is much improved. I have one question and a couple of recommendations:

I initially read the category of “non-employed” according to the US definition of unemployed. However, as it has been shown (Wilson & Walker, 1993, Voßemer, et al 2018) “unemployment has an adverse effect on health. This effect is still demonstrable when social class, poverty, age and pre-existing morbidity are adjusted for” Wilson & Walker 1993, p. 153). Which made me realize you non-employed may meaning something different. Can you please clarify.

Relabel Tables 3 and 4 to make clear the data define cluster 1 and cluster 2.

For some reason, I interpreted your title as focusing on positive health behaviors. But the labels of both your clusters emphasis negative practices which, although accurate, I initially interpreted as an error. My suggestion will be to change the label to emphasize the positive health practices in Cluster 1 (that you clearly list). “low risk, poor fruit and exercise with participants in it being characterized by the lowest probability for smoking, moderately low alcohol consumption, the highest vegetable intake, regular breakfast intake, as well as low fruit intake and physical activity.”

Wilson, S. H., & Walker, G. M. (1993). Unemployment and health: a review. Public health, 107(3), 153-162.

Voßemer, J., Gebel, M., Täht, K., Unt, M., Högberg, B., & Strandh, M. (2018). The effects of unemployment and insecure jobs on well-being and health: The moderating role of labor market policies. Social Indicators Research, 138(3), 1229-1257.

Reviewer #2: Thank you for your consideration of my comments in this revised manuscript. While I think that the edits based on my own and Reviewer 1’s comments have improved the paper, there are still a few areas where I think additional edits would further strengthen this.

In general, I think that the introduction is still slightly lacking with the justification of why this an important area to focus on. You state that it’s likely multiple health behaviours are interrelated, and that behaviour is related to happiness, but I think the introduction is missing a clear link between these two points. Why is it particularly important to look at clustered health behaviours specifically and how these are related to happiness?

I also think that the discussion could go further in depth when discussing the study findings as a whole. You state in your abstract that: “Identifying the clusters of multiple health behaviors and their effect on happiness can greatly contribute to developing strategies for enhancing happiness and improving health behaviors.” – I think that the discussion could go further when explaining and discussing this point.

My specific comments are outlined below:

1. The labelling of cluster 1 is still not quite right and doesn’t particularly make sense currently. In particular ‘poor fruit’ needs changing to something more descriptive, I would suggest ‘low risk: low fruit intake, low exercise’.

2. The first line of the introduction could be restructured to improve clarity of the point you are trying to make.

3. On a number of occasions throughout the manuscript, you use ‘I’ rather than sticking to third person (e.g. ‘this study investigated…) – these need editing into third person.

4. Page 6, line 111 – refers to ‘more recent ones’ – what does the ‘ones’ here refer to? Studies or behaviours? This needs clarifying.

5. The translated version of the alcohol item on p6, line 119 doesn’t make sense – I think this should read ‘how often do you consume alcohol’ or something similar

6. Line 113 refers to ‘enunciation’ when I think you mean ‘question’

7. Where does the 2 times daily fruit and vegetable cut off come from? In the UK, the standard healthy level of fruit and veg is 5 pieces of either fruit or veg per day so it would be useful to understand why this cut off was used.

8. Page 8, line 172, would it be possible to provide the USD equivalent to yen for the participant income levels? This would improve clarity for non-Japanese readers.

9. You mention that missing data was excluded so there was only a 31% response rate for participants, but I think you need to provide more information about the missing data – how much data was missing for each item/participant? Was there any consideration given to imputing this data rather than excluding participants with any data missing? While you state in your response to reviewer 1 that this is acknowledged in the limitations, I could not see this in the discussion and think that it needs to be clearly stated as a limitation of the study.

10. P10 refers to regular and non-regular employment, what does this mean and what are the definitions for these two types of employment? (e.g., does ‘regular’ employment include part time employment?)

11. The two categories of breakfast eating seem to overlap each other – ‘almost every day’ to me means that participants sometimes don’t eat breakfast which would mean they would also be in the ‘not every day’ category – this labelling needs clarifying.

12. Page 13-14 – the cluster 1 OR presented in the text which refers to Table 4 does not present the same OR as the figure then presented in Table 4.

13. By presenting the depression findings at the very beginning of the discussion, this seems to highlight these findings as the most important results from the study. I think this paragraph needs to be moved further down the discussion and instead the key findings from the study should be highlighted in the first paragraph of the discussion.

14. The first mention of cluster 1 and 2 in the discussion needs to remind the reader of what both of these clusters represent – I know that you have later defined these in the discussion, but I think it’s important to do this as soon as these are mentioned in the discussion section.

15. Page 16, You refer to the fact that younger participants having an unhealthier lifestyle as a health awareness issue – I think it also needs to be acknowledged here that this is likely to be a time constraint/societal issue rather than just being related to awareness.

7. PLOS authors have the option to publish the peer review history of their article (what does this mean?). If published, this will include your full peer review and any attached files.

Reviewer #1: No

Reviewer #2: No

---

## [Author Response · Author response to Decision Letter 1]

6 Jan 2022

Reviewer #1:

#1 I initially read the category of “non-employed” according to the US definition of unemployed. However, as it has been shown (Wilson & Walker, 1993, Voßemer, et al 2018) “unemployment has an adverse effect on health. This effect is still demonstrable when social class, poverty, age and pre-existing morbidity are adjusted for” Wilson & Walker 1993, p. 153). Which made me realize you non-employed may meaning something different. Can you please clarify.

Thank you for the comment. The “non-employed” category does mean “not unemployed or jobless,” but merely without occupation. Therefore, I added the annotation as follows:

non-employed; and without occupation, not jobless (p. 7, lines 167-168)

#2 Relabel Tables 3 and 4 to make clear the data define cluster 1 and cluster 2.

Thank you for the comment. I revised the labels for Clusters 1 and 2 in Tables 3 and 4 as follows:

Table 3: Association between health behavior cluster of “low-risk smoking and drinking, high vegetable intake, and regular breakfast” and socio-demographics

Table 4: Cluster 1: low-risk smoking and drinking, high vegetable intake, and regular breakfast; Cluster 2: high alcohol, poor nutrition, and exercise

#3 For some reason, I interpreted your title as focusing on positive health behaviors. But the labels of both your clusters emphasis negative practices which, although accurate, I initially interpreted as an error. My suggestion will be to change the label to emphasize the positive health practices in Cluster 1 (that you clearly list). “low risk, poor fruit and exercise with participants in it being characterized by the lowest probability for smoking, moderately low alcohol consumption, the highest vegetable intake, regular breakfast intake, as well as low fruit intake and physical activity.”

I appreciate the advice. Based on this insightful suggestion, I revised the description for Cluster 1 as follows:

“low-risk smoking and drinking, high vegetable intake, and regular breakfast,” (p. 11, line 224)

 

Reviewer #2: Thank you for your consideration of my comments in this revised manuscript. While I think that the edits based on my own and Reviewer 1’s comments have improved the paper, there are still a few areas where I think additional edits would further strengthen this.

In general, I think that the introduction is still slightly lacking with the justification of why this an important area to focus on. You state that it’s likely multiple health behaviours are interrelated, and that behaviour is related to happiness, but I think the introduction is missing a clear link between these two points. Why is it particularly important to look at clustered health behaviours specifically and how these are related to happiness?

I also think that the discussion could go further in depth when discussing the study findings as a whole. You state in your abstract that: “Identifying the clusters of multiple health behaviors and their effect on happiness can greatly contribute to developing strategies for enhancing happiness and improving health behaviors.” – I think that the discussion could go further when explaining and discussing this point.

Thank you for the comment. I revised the Introduction accordingly and added a brief implication for practice in the conclusion;

Scholars have found that healthy lifestyles, such as being physically active [16, 17], consuming vegetables and fruits [18-20], not smoking [21,22], and consuming alcohol in moderation [23-25], are significantly associated with happiness [26]. Thus, further engagement in health behaviors could enhance happiness [27]. Nonetheless, lifestyles are composed of multiple health behaviors. Thus, many studies have investigated the relationship between health behaviors and physical or psychological health status in isolation.

A few studies in Europe and the United States have found that multiple health behaviors tend to occur interrelatedly and that the degree of health consciousness exerts a cascading effect on all types of health behaviors [28-30]. In other words, one health behavior can trigger a chain reaction of other health behaviors, whereas the opposite is also true. Moreover, the simultaneous occurrence of multiple health behaviors has been attributed to various mechanisms, such as the commonality of multiple factors associated with such behaviors, and the fact that one behavior may be a coping strategy for another [31]. Multiple health behaviors exert synergistic effects. Therefore, research on the clustering of multiple health behaviors and their relationship with health outcomes is increasing [28,30,32-36].

Furthermore, scholars have analyzed the synergistic effects of multiple health behaviors on happiness [23, 37]. Specifically, the research on the association between multiple health behaviors and happiness has focused on younger populations of adolescents and college students [33-35]. However, less is known about multiple health behaviors across the full range of adult age population. Specifically, studies on this topic are lacking in Japan.

Intervention that targets only one health behavior can lead to consequences for improving other co-occurring health behaviors, which enhances happiness accordingly. Against this background, altering health behavior patterns could be an effective means for increasing happiness. Moreover, identifying the existing clusters of health behavior patterns and the characteristics of people belonging to these clusters would be beneficial to the formulation of intervention strategies for changing health behavior patterns. (pp. 3-4, l.59-1.84)

Based on the results, the synergistic effect of improving health behaviors can be efficiently induced by constructing measures with a compound and chain effect, instead of strategies that improve each health behavior. (p.19, l.396-398)

1. The labelling of cluster 1 is still not quite right and doesn’t particularly make sense currently. In particular ‘poor fruit’ needs changing to something more descriptive, I would suggest ‘low risk: low fruit intake, low exercise’.

Thank you for the advice. I added this following description for Cluster 1:

“low-risk smoking and drinking, high vegetable intake, and regular breakfast,” (p. 11, line 224)

2. The first line of the introduction could be restructured to improve clarity of the point you are trying to make.

Thank you for the comment. I revised the first line of the Introduction for a clearer meaning.

Happiness, which is a key concept of well-being, productivity, and quality of life, has received much scholarly attention worldwide. (p.3, l.46-l.47)

3. On a number of occasions throughout the manuscript, you use ‘I’ rather than sticking to third person (e.g. ‘this study investigated…) – these need editing into third person.

I appreciate your pointed out. I corrected first person to third person throughout the manuscript.

4. Page 6, line 111 – refers to ‘more recent ones’ – what does the ‘ones’ here refer to? Studies or behaviours? This needs clarifying.

Thank you for pointing out this aspect. I modified the first-person voice to that of the third person throughout the manuscript. “Ones” refers to studies. I have revised it from “more recent ones” to “recent studies” (p. 5, line 109)

5. The translated version of the alcohol item on p6, line 119 doesn’t make sense – I think this should read ‘how often do you consume alcohol’ or something similar.

Thank you for the comment. I revised the question about alcohol intake as follows:

“How frequently do you consume alcohol?” (p. 5, lines 116-117)

6. Line 113 refers to ‘enunciation’ when I think you mean ‘question.’

Thank you for pointing out this aspect. I corrected the word to “question” (p. 6, line 127)

7. Where does the 2 times daily fruit and vegetable cut off come from? In the UK, the standard healthy level of fruit and veg is 5 pieces of either fruit or veg per day so it would be useful to understand why this cut off was used.

Thank you for the comment. I added a note about the cut-off value for fruit and vegetable intake and the reference.

Consumption was measured with reference to the recommendation by the Ministry of Health, Labor and Welfare to eat at least two meals per day with side dishes including vegetables and fruits or an intake of at least 2SV per day of fruits and vegetables [40, 41]. (p. 6, lines 131-134)

8. Page 8, line 172, would it be possible to provide the USD equivalent to yen for the participant income levels? This would improve clarity for non-Japanese readers.

Thank you for the comment. As the question used only JPY, I could not provide the categories representing USD. Therefore, I provided the exchange rate between USD and JPY at the time of the survey.

SES (i.e., expressed in increments of 10,000 JPY; 1 USD was approximately 113 JPY at the time of the survey), (p. 7, lines 164-165)

9. You mention that missing data was excluded so there was only a 31% response rate for participants, but I think you need to provide more information about the missing data – how much data was missing for each item/participant? Was there any consideration given to imputing this data rather than excluding participants with any data missing? While you state in your response to reviewer 1 that this is acknowledged in the limitations, I could not see this in the discussion and think that it needs to be clearly stated as a limitation of the study.

Thank you for the comment. I imputed the missing data for happiness and depression with mean. However, health behaviors (categorical variables) and socio-demographic attributes (employment status and academic background) could not be imputed. Thus, responses with missing data on health behaviors, employment status, and academic background were excluded. The characteristics of the excluded participants were added to the Results section. Moreover, I cited the possibility of sample bias in the Limitations section as follows:

Participants with missing data on the socio-demographic attributes (employment status and academic background) and health behaviors were excluded. The average age of the participants with missing data was 53.98 ± 11.23 years contrary to 49.84 ± 9.16 years of the analyzed sample (t = −11.60, p < 0.000), whereas 59.2% contrary to 59.5% were male in the analyzed sample (χ2 = 114.18, p < 0.000). (p. 9, lines 201-205)

Moreover, differences in age and sex distribution were observed between the included participants and sample with missing data. (p. 18, lines 370-371)

10. P10 refers to regular and non-regular employment, what does this mean and what are the definitions for these two types of employment? (e.g., does ‘regular’ employment include part time employment?)

Thank you for the comment. I added the annotation about employment status as follows:

(i.e., regular; full-time with a contract for regular employment; non-regular; part-time or temporary contract; self-employed; or non-employed; without an occupation, not jobless). (p. 7, lines 166-168)

11. The two categories of breakfast eating seem to overlap each other – ‘almost every day’ to me means that participants sometimes don’t eat breakfast which would mean they would also be in the ‘not every day’ category – this labelling needs clarifying.

Thank you for the comment. I grouped the responses of breakfast-eating habit into “almost every day” and “skip 2−3 times/week.” Subsequently, I revised the description for the response category about breakfast-eating habit as follows:

Responses were coded 0 for “less than 4 days/week” and 1 for “more than 4 days/week.” (p. 6, lines 137-138)

12. Page 13-14 – the cluster 1 OR presented in the text which refers to Table 4 does not present the same OR as the figure then presented in Table 4.

Thank you for this comment. I corrected the OR and CI for Cluster 1 as follows:

OR = 1.266, 95% CI = 1.010−1.587, p = 0.041 (p. 12, line 246)

13. By presenting the depression findings at the very beginning of the discussion, this seems to highlight these findings as the most important results from the study. I think this paragraph needs to be moved further down the discussion and instead the key findings from the study should be highlighted in the first paragraph of the discussion.

I appreciate your pointed out. Based on your suggestion, I moved the depression findings to the end of the discussion.

Meanwhile, this study reported that 33% of the participants had depressive tendencies. In a recent study, approximately 25% of Japanese adults suffer from mental health problems, such as anxiety disorders and mood disorders [94]. Therefore, although the prevalence of depression in the samples of the present study was slightly higher than that for mental health problems in the Japanese population, it did not deviate significantly from the trend. (p. 17, lines 357-361)

14. The first mention of cluster 1 and 2 in the discussion needs to remind the reader of what both of these clusters represent – I know that you have later defined these in the discussion, but I think it’s important to do this as soon as these are mentioned in the discussion section.

Thank you for this comment. Based on the suggestion, I added the definitions for Clusters 1 and 2 in the first part of the Discussion section.

Based on the distribution of health behavior, the participants in Cluster 1 (low-risk smoking and drinking, high vegetable intake, and regular breakfast) exhibited low health risk behaviors than those in Cluster 2 (high alcohol, poor nutrition, and exercise) (p. 19, lines 396-398)

15. Page 16, You refer to the fact that younger participants having an unhealthier lifestyle as a health awareness issue – I think it also needs to be acknowledged here that this is likely to be a time constraint/societal issue rather than just being related to awareness.

Thank you for the comment. I added a discussion on the tendency of the younger participants toward an unhealthy lifestyle as follows:

Nevertheless, other surveys have found that people aged in their 30s and 40s tend to be more constrained in terms of time and are less aware of their health due to increased roles and responsibilities in family and work life [41,62], (p. 15, lines 287-289)

---

## [Decision Letter · Decision Letter 2]

23 Feb 2022

PONE-D-21-10430R2Clustering of health behaviors among Japanese adults and their association with socio-demographics and happinessPLOS ONE

Dear Dr. Satoh,

Thank you for submitting your manuscript to PLOS ONE. After careful consideration, we feel that it has merit but does not fully meet PLOS ONE’s publication criteria as it currently stands. Therefore, we invite you to submit a revised version of the manuscript that addresses the points raised during the review process.

We look forward to receiving your revised manuscript.

Kind regards,

Akihiro Nishi, M.D., Dr.P.H.

Academic Editor

PLOS ONE

Journal Requirements:

Additional Editor Comments (if provided):

Please do the final revision before accept.

Reviewers' comments:

Reviewer's Responses to Questions

**Comments to the Author**

1. If the authors have adequately addressed your comments raised in a previous round of review and you feel that this manuscript is now acceptable for publication, you may indicate that here to bypass the “Comments to the Author” section, enter your conflict of interest statement in the “Confidential to Editor” section, and submit your "Accept" recommendation.

Reviewer #1: (No Response)

Reviewer #2: All comments have been addressed

2. Is the manuscript technically sound, and do the data support the conclusions?

Reviewer #1: Yes

Reviewer #2: Yes

3. Has the statistical analysis been performed appropriately and rigorously? 

Reviewer #1: Yes

Reviewer #2: Yes

4. Have the authors made all data underlying the findings in their manuscript fully available?

Reviewer #1: Yes

Reviewer #2: No

5. Is the manuscript presented in an intelligible fashion and written in standard English?

Reviewer #1: No

Reviewer #2: Yes

6. Review Comments to the Author

Reviewer #1: While the author responded to the reviewers’ comments and provide clarification, significant issues remain.

Line 131 Make it clear if the classification is for current smokers vs non-smokers or never smokers vs ever smoker.

Line 138 spell out 2SV

Line 164 “Cut-off scores of 3 and 3 were used for depression.” Clarify

line 171-173 your clarification regarding employment status does not solve the problem. What does without an occupation mean? meaning out of the labor force? Provide a clear definition, I found these two articles very informative and wondered if that is what you were talking about.

Gnambs, T., Stiglbauer, B., & Selenko, E. (2015). Psychological effects of (non) employment: A cross‐national comparison of the United States and Japan. Scandinavian Journal of Psychology, 56(6), 659-669.

Genda, Y. (2007). Jobless youths and the NEET problem in Japan. Social Science Japan Journal, 10(1), 23-40.

Line 222 - What about labeling your clusters something brief but informative and consistent, rather than Cluster 1 and Cluster 2. Your inconsistent use of descriptors to show directionality for both healthy (low fruit intake) and unhealthy (low risk smoking) behaviors makes it harder to keep the direction of the findings in order.

Even your conclusion is confusing: Thus, belonging to the cluster of low-risk health behaviors (low fruit intake, low exercise) may be significantly linked to high levels of happiness. What is the role of “low-risk smoking and drinking, high vegetable intake, and regular breakfast”?

Line 230 should be lower

Line 236 “regular employees (OR = 0.540, 95% CI = 0.341-0.857, p = 0.009), non-regular employees (OR = 0.576, 95% CI = 0.368-0.903, p = 0.016), and self-employed individuals (OR = 0.539, 95% CI = 0.322-0.901, p = 0.018) displayed lower odds of belonging to Cluster 1.” Does it mean that working people engage in poor health practices? Say it clearly in the discussion, there is a subtle reference only.

Line 255 “However, those in Cluster 1 were not entirely healthy” Actually you do not know how healthy they are, just know about their healthy practices. So maybe something like: “However, even individuals engage in "health living" (cluster 1), did not embrace all health recommendations equally.”

Line 261 Replace “findings” for “pattern”

Line 272 “The participants in Cluster 2 also exhibited moderate smoking” What does moderate smoking mean?!! Smoking is smoking!

Lines 285 – 286 Reword older ones vs younger ones, poorly worded

Line 287 – 289 State it clearly. Is this the reason why working people are more likely to be in cluster 2?

Line 306 – 309 – One can also speculate, higher education, higher income, more resources, more free time …

Line 329 “many studies” but cite only 1 – reference 87

Line 357 replace “reported” with found

Reviewer #2: Thank you for addressing my comments in this revision. I have one further (very minor) suggestion which is that on P3, line 46 I would replace the word 'concept' with 'facet'.

7. PLOS authors have the option to publish the peer review history of their article (what does this mean?). If published, this will include your full peer review and any attached files.

Reviewer #1: No

Reviewer #2: No

---

## [Author Response · Author response to Decision Letter 2]

7 Mar 2022

Review Comments to the Author

Reviewer #1: While the author responded to the reviewers’ comments and provide clarification, significant issues remain.

#1

• Line 131 Make it clear if the classification is for current smokers vs non-smokers or never smokers vs ever smoker.

Response: Thank you for your suggestion. I have included additional information about grouping of smoking status as follows:

Responses were coded as 1 for current non-smokers (responded “I have quit smoking” or “Never”) and 0 for current smokers (responded “I smoke” or “I smoke sometimes”). (p. 7, lines 132–133)

#2

• Line 138 spell out 2SV

Response: Thank you for your comment. I have spelled 2SV out to two servings (p. 6, line 140) 

#3

• Line 164 “Cut-off scores of 3 and 3 were used for depression.” Clarify

Response: Thank you for your comment. Based on your comment, I have added information about cut-off point of the GHQ

The cut-off point of the GHQ was set at 2/3 [46,47]; those scoring 3 points or higher were considered to indicate depression. (p. 8, lines 165–167)

#4

• Line 171-173 your clarification regarding employment status does not solve the problem. What does without an occupation mean? meaning out of the labor force? Provide a clear definition, I found these two articles very informative and wondered if that is what you were talking about.

Gnambs, T., Stiglbauer, B., & Selenko, E. (2015). Psychological effects of (non) employment: A cross‐national comparison of the United States and Japan. Scandinavian Journal of Psychology, 56(6), 659-669.

Genda, Y. (2007). Jobless youths and the NEET problem in Japan. Social Science Japan Journal, 10(1), 23-40.

Response: Thank you for your comment and for suggesting relevant literature. I have revised “without an occupation” to “non-employed” based on the recommended literature.

(… non-employed: those who are neither workers nor unemployed, such as full-time housewives, students or retirees) [48, 49]. (pp. 7-8, lines 175–176)

#5

• Line 222 - What about labeling your clusters something brief but informative and consistent, rather than Cluster 1 and Cluster 2. Your inconsistent use of descriptors to show directionality for both healthy (low fruit intake) and unhealthy (low risk smoking) behaviors makes it harder to keep the direction of the findings in order.

Even your conclusion is confusing: Thus, belonging to the cluster of low-risk health behaviors (low fruit intake, low exercise) may be significantly linked to high levels of happiness. What is the role of “low-risk smoking and drinking, high vegetable intake, and regular breakfast”?

Response: Thank you for your suggestion. I agree with you and have attempted to revise the labels of Clusters 1 and 2 to reflect their characteristics to the best of my ability. I have also revised the discussion of the characteristics of both clusters. Accordingly, I have used the new labels instead of Cluster 1 and Cluster 2 after the result section.

Cluster 1 (n = 817; 52.3%) was characterized by the lowest probability for smoking, moderately low alcohol consumption, highest vegetable intake, and regular breakfast intake and was described as “low substance use and good dietary habit.” Cluster 2 (n = 737; 47.7%) was described as “high alcohol, poor nutrition, and inactive,” with participants characterized by high levels of alcohol consumption, low vegetable intake, very low fruit intake, and very low physical activity. (p. 12, lines 225–231)

Based on the distribution of health behaviors, participants in the “low substance use and good dietary habit” cluster exhibited low health risk behaviors than those in the “low substance use and good dietary habit” cluster. Having breakfast regularly may be associated with high vegetable consumption [51-53]. Previous studies have suggested that those who show high health awareness are likely to have some healthy behaviors simultaneously [37,54–57]. Thus, the “low substance use and good dietary habit” cluster, comprising individuals who are non-smokers, moderate drinkers of alcohol, eat vegetables nearly every day, eat breakfast nearly every day, even though with low fruit intake, and are physically inactive, was considered a health behavior pattern. (p. 15, lines 258–267)

#6

• Line 230 should be lower

Response: Thank you for your pointing this out. I replaced “low” with “lower.” (p. 12, line 231)

#7

• Line 236 “regular employees (OR = 0.540, 95% CI = 0.341-0.857, p = 0.009), non-regular employees (OR = 0.576, 95% CI = 0.368-0.903, p = 0.016), and self-employed individuals (OR = 0.539, 95% CI = 0.322-0.901, p = 0.018) displayed lower odds of belonging to Cluster 1.” Does it mean that working people engage in poor health practices? Say it clearly in the discussion, there is a subtle reference only.

Response: Thank you for your comment. As you pointed out, this study found that working people could have poor health practices. Therefore, I have revised the discussion on the association between clusters of health behaviors and employment status. 

Additionally, other surveys have found that people in their 30s–40s tend to face time constraints and are less aware of their health due to increased roles and responsibilities in family and work life [42,71]. That could explain the lack of association between younger age group and the “low substance use and good dietary habit” cluster and it might also indicate that employed people were more likely to be in the “high alcohol, poor nutrition, and inactive” cluster. The present study demonstrated that the employed Japanese population was more prone to unhealthy behaviors. A large proportion of the insured, working adults failed to meet the recommendations for health behaviors in the US [72]. Other studies have also reported that many employed people tend to observe unhealthy lifestyles, because they are extremely busy with work and fail to pay attention to their health habits [42,73,74]. Moreover, work-related stress may influence engagement in health risk behaviors, such as alcohol consumption [75,76]. The work culture may also lead employed individuals to frequently go out for drinks to interact socially and professionally with bosses and colleagues after work [77,78]. Hence, employed individuals in Japan may be more likely to adopt unhealthy behaviors, which highlights the potential need for a comprehensive analysis of the work lives of Japanese workers to improve health behavior patterns. (p.17, lines 307–321)

#8

• Line 255 “However, those in Cluster 1 were not entirely healthy” Actually you do not know how healthy they are, just know about their healthy practices. So maybe something like: “However, even individuals engage in "health living" (cluster 1), did not embrace all health recommendations equally.”

Response: Thank you for your pointing it out. Based on your advice, I have revised as follows:

However, individuals in this cluster that engaged in “health living” did not embrace all health recommendations equally. (p. 15, lines 266–267)

#9

• Line 261 Replace “findings” for “pattern”

Response: Thank you for your pointing it out. I replaced “findings” with “pattern.” (p. 15, line 269)

#10

• Line 272 “The participants in Cluster 2 also exhibited moderate smoking” What does moderate smoking mean?!! Smoking is smoking!

Response: Thank you for your insightful comment. I revised here as follows:

“High alcohol, poor nutrition, and inactive” cluster also exhibited higher current smoking habit and lower vegetable intake than those in the “low substance use and good dietary habit” cluster. (p. 16, lines 283–285)

#11

• Lines 285 – 286 Reword older ones vs younger ones, poorly worded

Response: Thank you for your insightful comment. I reworded these words to be as appropriate as possible.

Japanese people in their 50s–60s tend to display increased levels of health awareness and adopt healthier lifestyles compared with the people in their 30s–40s [42,66]. (p. 16, lines 298–299)

#12

• Line 287 – 289 State it clearly. Is this the reason why working people are more likely to be in cluster 2?

Response: Thank you for your comment. Related to comment #7, this study found that working individuals could follow poor health practices, and I speculated the reasons for this as follows: 

Additionally, other surveys have found that people in their 30s–40s tend to be face time constraints and are less aware of their health due to increased roles and responsibilities in family and work life [42,71]. That could explain the lack of association between younger age group and “low substance use and good dietary habit” cluster, and it might also indicate that employed people were more likely to be in “high alcohol, poor nutrition, and inactive” cluster. The present study demonstrated that the employed Japanese population was more prone to unhealthy behaviors. A large proportion of the working insured adults failed to meet the recommendations for health behaviors in the US [72]. Other studies have also reported that many employed people tend to observe unhealthy lifestyles, because they are extremely busy with work and fail to pay attention to their health habits [42,73,74]. Moreover, work-related stress may influence engagement in health risk behaviors, such as alcohol consumption [75,76]. The work culture may also lead employed individuals to frequently go out for drinks to interact socially and professionally with bosses and colleagues after work [77,78]. Hence, employed individuals in Japan may be more likely to adopt unhealthy behaviors, which highlights the potential need for a comprehensive analysis of the work lives of Japanese workers to improve health behavior patterns. (p.17, lines 307–321)

#13

• Line 306 – 309 – One can also speculate, higher education, higher income, more resources, more free time …

Response: Thank you for your insightful comment. I agree with your comment that those with higher level of education or higher income could have more resources for health promotion. However, it was unclear whether such people could have free time to practice healthy habits. Therefore, I have revised the text from the standpoint of education level and SES as follows:

According to previous studies, high levels of education may be significantly associated with health behaviors [30,83], which is similar to the results obtained by the current study. In Japan, the level of education is related to health behaviors, where high levels of education are associated with more physical exercise routines [84], lower problem drinking [85], and lower obesity [86]. In addition, the current study confirmed the association between multiple healthy behaviors and the higher SES. This result was also congruent with literature [80,87]. Scholars have speculated that this finding is due to the fact that the higher the level of education or the higher income, the more knowledge or resources people utilize for health [88,89]. Therefore, they are more likely they are to engage in health-promotion activities and also gain access to useful resources that can improve health. This indicates the need to examine the content of education in schools to ensure that young people, regardless of their SES, gain appropriate knowledge about health and health behaviors from the stage of primary education. (pp. 17–18, lines 327–338)

#14

• Line 329 “many studies” but cite only 1 – reference 87

Response: Thank you for your pointing out. I apologize for the confusion. I have revised accordingly, as follows.

One systematic review and metanalysis on health behavior patterns reported no sex differences in health risk behaviors [90] (p. 18, lines 339–340)

#15

• Line 357 replace “reported” with found

Response: Thank you for pointing this out. I replaced “reported” with “found.” (p. 19, line 367)

Reviewer #2

• Thank you for addressing my comments in this revision. I have one further (very minor) suggestion which is that on P3, line 46 I would replace the word ‘concept’ with ‘facet’.

Response: Thank you for your advice. I have replaced “concept” with “facet.” (p. 4, line 47)

---

## [Editor Report · Decision Letter 3]

14 Mar 2022

Clustering of health behaviors among Japanese adults and their association with socio-demographics and happiness

PONE-D-21-10430R3

Dear Dr. Satoh,

We’re pleased to inform you that your manuscript has been judged scientifically suitable for publication and will be formally accepted for publication once it meets all outstanding technical requirements.

Kind regards,

Akihiro Nishi, M.D., Dr.P.H.

Academic Editor

PLOS ONE

Additional Editor Comments (optional):

This handling editor is happy to see the authors addressed the comments on the minor revision and to accept the manuscript.
---

## [Editor Report · Acceptance letter]

6 Apr 2022

PONE-D-21-10430R3 

Clustering of health behaviors among Japanese adults and their association with socio-demographics and happiness 

Dear Dr. Satoh:

I'm pleased to inform you that your manuscript has been deemed suitable for publication in PLOS ONE. Congratulations! Your manuscript is now with our production department. 

Kind regards, 

on behalf of

Dr. Akihiro Nishi 

Academic Editor

PLOS ONE